# CONJNORM: TRACTABLE DENSITY ESTIMATION FOR OUT-OF-DISTRIBUTION DETECTION

**Bo Peng**[1]\* **Yadan Luo**[2]\* **Yonggang Zhang**[3], **Yixuan Li**[4], **Zhen Fang**[1]†
University of Technology Sydney, Australia[1]
The University of Queensland, Australia[2]
Hong Kong Baptist University, Hong Kong[3]
University of Wisconsin-Madison, USA[4]
`bo.peng-7@student.uts.edu.au, y.luo@uq.edu.au`
`csygzhang@comp.hkbu.edu.hk, sharonli@cs.wisc.edu`
`zhen.fang@uts.edu.au`

## ABSTRACT

Post-hoc out-of-distribution (OOD) detection has garnered intensive attention in reliable machine learning. Many efforts have been dedicated to deriving score functions based on logits, distances, or rigorous data distribution assumptions to identify low-scoring OOD samples. Nevertheless, these estimate scores may fail to accurately reflect the true data density or impose impractical constraints. To provide a unified perspective on density-based score design, we propose a novel theoretical framework grounded in Bregman divergence, which extends distribution considerations to encompass an exponential family of distributions. Leveraging the conjugation constraint revealed in our theorem, we introduce a CONJNORM method, reframing density function design as a search for the optimal norm coefficient $p$ against the given dataset. In light of the computational challenges of normalization, we devise an unbiased and analytically tractable estimator of the partition function using the Monte Carlo-based importance sampling technique. Extensive experiments across OOD detection benchmarks empirically demonstrate that our proposed CONJNORM has established a new state-of-the-art in a variety of OOD detection setups, outperforming the current best method by up to $13.25\%$ and $28.19\%$ (FPR95) on CIFAR-100 and ImageNet-1K, respectively.

## 1 INTRODUCTION

Despite the significant progress in machine learning that has facilitated a broad spectrum of classification tasks (Gaikwad et al., 2010; Huang et al., 2014; Zhao et al., 2019; Shantaiya et al., 2013; Krizhevsky et al., 2012; Masana et al., 2022), models often operate under a *closed-world* scenario, where test data stems from the same distribution as the training data. However, real-world applications often entail scenarios in which deployed models may encounter unseen classes of samples during training, giving rise to what is known as out-of-distribution (OOD) data. These OOD instances have the potential to undermine a model's stability and, in certain cases, inflict severe damage upon its performance. To identify and safely remove these OOD data in decision-critical tasks (Chen et al., 2022; Zimmerer et al., 2022), OOD detection techniques have been proposed. To facilitate easy separation of in-distribution (ID) and OOD data, mainstream OOD approaches either leverage post-hoc analysis or model re-training (Ming et al., 2022a; Wei et al., 2022; Chen et al., 2021; Huang & Li, 2021; Du et al., 2022; Katz-Samuels et al., 2022; Wang et al., 2023; Lee et al., 2017) by using density-based (Morteza & Li, 2022), output-based (Liu et al., 2020), distance-based (Lee et al., 2018) and reconstruction-based strategies (Zhou, 2022).

Following previous works (Liu et al., 2020; Liang et al., 2017; Hendrycks & Gimpel, 2016; Sun et al., 2021; Ahn et al., 2023; Djurisic et al., 2022; Lee et al., 2018), we focus on the *post-hoc OOD detection* strategy, which offers more practical advantages than learning-based OOD approaches

---

\*Equal Contribution
†Correspondence to Zhen Fang (zhen.fang@uts.edu.au)

without requiring resource-intensive re-training processes. Our key research question of this approach centers on how to derive a proper *scoring functions* to indicate the ID-ness of the input for effectively discerning OOD samples during testing. By definition, OOD data inherently diverges from ID data by means of their data density distributions, rendering *estimated density* an ideal metric for discrimination. Nevertheless, it is *non-trivial* to parameterize the unknown ID data distribution for density estimation since the computation of normalization constants tends to be costly and even intractable (Gutmann & Hyvärinen, 2012a). While recent attempts have been made by modeling ID data as some specific prior distributions, *i.e.*, the Gibbs-Boltzmann distribution in (Liu et al., 2020) and the mixture Gaussian distribution in (Morteza & Li, 2022), to factitiously make normalization constants sample-independent or known, this practice imposes strong distributional assumptions on the underlying feature space. Furthermore, it offers no theoretical guarantee that those pre-defined distributions necessarily hold in practice.

In this paper, we introduce an innovative Bregman divergence-based (Banerjee et al., 2005) theoretical framework aimed at providing a unified perspective for designing density functions within an expansive exponential family of distributions (Amari, 2016). This framework not only bridges the gap between existing post-hoc OOD approaches (Liu et al., 2020; Morteza & Li, 2022) but also highlights a valuable conjugation constraint for tailoring density functions to given datasets. Without loss of generality, we focus on the conjugate pair of $l_p$ and $l_q$ norms and propose the CON-JNORM method. This approach reframes the density function design as a search for the optimal norm coefficient within a narrow range. To facilitate tractable estimation of the partition function for normalization, we compare two existing estimation baselines and put forward a Monte Carlo-based importance sampling technique, which yields an unbiased and analytically tractable estimator.

## 2 PRELIMINARIES

Let $\mathcal{X}$ and $\mathcal{Y} = \{1, \ldots, K\}$ represent the input space and ID label space, respectively. The joint ID distribution, represented as $P_{X_I Y_I}$, is a joint distribution defined over $\mathcal{X} \times \mathcal{Y}$. During testing time, there are some unknown OOD joint distributions $D_{X_O Y_O}$ defined over $\mathcal{X} \times \mathcal{Y}^c$, where $\mathcal{Y}^c$ is the complementary set of $\mathcal{Y}$. We also denote $p_I(\mathbf{x})$ as the density of the ID marginal distribution $P_{X_I}$. According to (Fang et al., 2022), OOD detection can be formally defined as follows:

**Problem 1** (OOD Detection). *Given labelled ID data $\mathcal{D}_{in} = \{(\mathbf{x}_1, \mathbf{y}_1), ..., (\mathbf{x}_N, \mathbf{y}_N)\}$, which is drawn from $P_{X_I Y_I}$ independent and identically distributed, the aim of OOD detection is to learn a predictor $g$ by using $\mathcal{D}_{in}$ such that for any test data $\mathbf{x}$: 1) if $\mathbf{x}$ is drawn from $D_{X_I}$, then $g$ can classify $\mathbf{x}$ into correct ID classes, and 2) if $\mathbf{x}$ is drawn from $D_{X_O}$, then $g$ can detect $\mathbf{x}$ as OOD data.*

**Post-hoc Detection Strategy.** Many representative OOD detection methods (Liu et al., 2020; Liang et al., 2017; Hendrycks & Gimpel, 2016; Sun et al., 2021; Ahn et al., 2023; Djurisic et al., 2022; Lee et al., 2018) follow a post-hoc strategy, *i.e.,* given a well-trained model $\mathbf{f}_{\boldsymbol{\theta}}$ using $\mathcal{D}_{in}$, and a scoring function $S$, then $\mathbf{x}$ is detected as ID data if and only if $S(\mathbf{x}; \mathbf{f}_{\boldsymbol{\theta}}) \geq \lambda$, for some given threshold $\lambda$:

$$g(\mathbf{x}) = \text{ID, if } S(\mathbf{x}; \mathbf{f}_{\boldsymbol{\theta}}) \geq \lambda; \text{ otherwise, } g(\mathbf{x}) = \text{OOD.} \quad (1)$$

Following the representative work (Morteza & Li, 2022), a natural view for the motivation of the post-hoc strategy is to use a level set for ID density $p_I(\mathbf{x})$ to discern ID and OOD data. Its main objective is to construct an efficient scoring function $S$, that can effectively replicate the behavior of the ID density function, $p_I(\mathbf{x})$, *i.e.*, $S(\mathbf{x}; \mathbf{f}_{\boldsymbol{\theta}}) \propto p_I(\mathbf{x})$. Therefore, using the density-based framework, (Morteza & Li, 2022) rewrites the post-hoc strategy as follows: given ID data density function $\hat{p}_{\boldsymbol{\theta}}(\cdot)$ estimated by well-trained model $\mathbf{f}_{\boldsymbol{\theta}}$ and a pre-defined threshold $\lambda$, then for any data $\mathbf{x} \in \mathcal{X}$,

$$g(\mathbf{x}) = \text{ID, if } \hat{p}_{\boldsymbol{\theta}}(\mathbf{x}) \geq \lambda; \text{ otherwise, } g(\mathbf{x}) = \text{OOD.} \quad (2)$$

In this work, we mainly utilize the density-based framework to design our theory and algorithm.

**Density Estimation Modeling.** The performance of density-based OOD detection heavily relies on the alignment between the estimated data density $\hat{p}_{\boldsymbol{\theta}}(\mathbf{x})$ and the true density $p_I(\mathbf{x})$. Considering a commonly used assumption in OOD detection, *i.e.,* the uniform class prior on ID classes (Jiang et al., 2023), $\hat{p}_{\boldsymbol{\theta}}(\mathbf{x})$ can be expressed as the aggregate of the ID class-conditioned distributions $\hat{p}_{\boldsymbol{\theta}}(\mathbf{x}|k)$:

$$\hat{p}_{\boldsymbol{\theta}}(\mathbf{x}) = \sum_{k=1}^{K} \hat{p}_{\boldsymbol{\theta}}(\mathbf{x}|k) \cdot \hat{p}_{\boldsymbol{\theta}}(k) \propto \sum_{k=1}^{K} \hat{p}_{\boldsymbol{\theta}}(\mathbf{x}|k). \quad (3)$$

Based on Eq. 3, our main objective is to estimate the class-conditional distribution of ID data, in order to effectively construct the data density $\hat{p}_{\boldsymbol{\theta}}$ for discriminating between ID and OOD data.

Without loss of generality, we employ latent features $\mathbf{z}$ extracted from deep models as a surrogate for the original high-dimensional raw data $\mathbf{x}$. This is because $\mathbf{z}$ is deterministic within the post-hoc framework. Consistent with probabilistic theory, we express $\hat{p}_{\boldsymbol{\theta}}(\mathbf{z}|k)$ in the following general form:

$$\hat{p}_{\boldsymbol{\theta}}\left(\mathbf{z}|k\right) = \frac{g_{\boldsymbol{\theta}}(\mathbf{z}, k)}{\Phi(k)}, \tag{4}$$

where $g_{\boldsymbol{\theta}}(\mathbf{z}, k)$ represents a non-negative density function, and $\Phi(k) = \int g_{\boldsymbol{\theta}}(\mathbf{z}, k) \, d\mathbf{z}$ denotes the partition function for normalization. According to prior works, the design principle for $g_{\boldsymbol{\theta}}(\mathbf{z}, k)$ can be divided into 3 categories: *logit-based*, *distance-based* and *density-based* methods.

**Logit-based OOD methods** (Liu et al., 2020; Hendrycks et al., 2019) resort to derive $g_{\boldsymbol{\theta}}(\mathbf{z}, k)$ from logit outputs. As a representative work, energy-based method (Liu et al., 2020) explicitly acknowledges $g_{\boldsymbol{\theta}}(\mathbf{z}, k)$ by fitting to the Gibbs-Boltzmann distribution, *i.e.*, $g_{\boldsymbol{\theta}}(\mathbf{z}, k) = \exp(f_{\boldsymbol{\theta}}^{k}/T)$ where $f_{\boldsymbol{\theta}}^{k}$ is the kth coordinate of $\mathbf{f}_{\boldsymbol{\theta}}$ and $T$ is a temperature parameter. This directly results in an energy-based scoring function $E(\mathbf{z}) = -T \log \sum_{k=1}^{K} g_{\theta}(\mathbf{z}, k)$. However, it can be easily checked from Eq. 4 that $E(\mathbf{z}) \propto -\log p(\mathbf{z})$ holds if and only if $\Phi(k) = \text{constant}, \forall k \in \mathcal{Y}$. While the energy-based method has demonstrated empirical effectiveness, it is essential to recognize that this condition, *i.e.*, $\Phi(k) = \text{constant}, \forall k \in \mathcal{Y}$, may not always hold in practical scenarios. Differently, Hendrycks & Gimpel (2016) proposes the maximum softmax score (MSP) to estimate OOD uncertainty:

$$\text{MSP}(\mathbf{z}) = \max_{k=1,\ldots,K} \hat{p}_{\boldsymbol{\theta}}(k|\mathbf{z}) = \max_{k=1,\ldots,K} \frac{g_{\theta}(\mathbf{z}, k)}{\sum_{k'=1}^{K} g_{\theta}(\mathbf{z}, k')} \not\propto \hat{p}_{\boldsymbol{\theta}}(\mathbf{z}). \tag{5}$$

where $g_{\theta}(\mathbf{z}, k) = \exp(f_{\boldsymbol{\theta}}^{k})$. However, as shown in Eq. 5, there exists a misalignment between MSP and the true data density, making ultimately MSP a suboptimal solution to OOD detection.

**Distance-based OOD methods** (Lee et al., 2017) target on deriving $g_{\boldsymbol{\theta}}(\mathbf{z}, k)$ by assessing the proximity of the input to the $k$-th prototype $\mu_k$. The selection of appropriate similarity metrics is crucial in capturing the intrinsic geometric data relationships. One of the most representative metrics used is the maximum Mahalanobis distance (Lee et al., 2017), which is formally defined as,

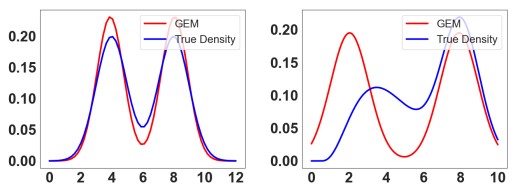

Figure 1: Illustration of the alignment of GEM score and true density of Gaussian (Left) and Gamma (Right) distributions.

$$\begin{aligned} \text{Maha}(\mathbf{z}) &= \max_{k=1,\ldots,K} -(\mathbf{z} - \boldsymbol{\mu}_k)^{\top} \Sigma^{-1} (\mathbf{z} - \boldsymbol{\mu}_k) \\ &= \max_{k=1,\ldots,K} \log g_{\theta}(\mathbf{z}, k) \not\propto \hat{p}_{\boldsymbol{\theta}}(\mathbf{z}). \end{aligned}$$

The distance metric can be considered as the density function $g_{\theta}(\mathbf{z}, k)$ in Eq. 4. This interpretation allows us to bypass the estimation of the partition function and leads to a significant observation: the distance measures are not directly proportional to the true data density.

**Density-based OOD methods** have rarely been studied compared to the previous two groups, primarily because of the complexities involved in estimating $\Phi(k)$. Recently, a method called GEM (Morteza & Li, 2022) has been proposed, with the assumption that the class-conditional density conforms to a Gaussian distribution: let $g_{\boldsymbol{\theta}}(\mathbf{z}, k) = \exp(-\frac{1}{2}(\mathbf{z} - \boldsymbol{\mu}_k)^{\top} \Sigma^{-1}(\mathbf{z} - \boldsymbol{\mu}_k))$,

$$\text{GEM}(\mathbf{z}) = \sum_{k=1}^{K} \frac{\exp(-\frac{1}{2}(\mathbf{z} - \boldsymbol{\mu}_k)^{\top} \Sigma^{-1}(\mathbf{z} - \boldsymbol{\mu}_k))}{\sqrt{(2\pi)^d |\Sigma|}} = \sum_{k=1}^{K} \frac{g_{\boldsymbol{\theta}}(\mathbf{z}, k)}{\Phi(k)} \propto \frac{1}{K} \sum_{k=1}^{K} \hat{p}_{\boldsymbol{\theta}}(\mathbf{z}|k) = \hat{p}_{\boldsymbol{\theta}}(\mathbf{z}),$$

where $\Sigma \in \mathbb{R}^{d \times d}$ is the covariance matrix. Note that $\Phi(k) = \sqrt{(2\pi)^d |\Sigma|}$ in this case. This Gaussian assumption, while simplifying the estimation of $\Phi(k)$, enables the direct utilization of the Mahalanobis distance as $g_{\boldsymbol{\theta}}(\mathbf{z}, k)$. However, it is crucial to acknowledge that this methodology may impose constraints on its ability to generalize effectively across a wide range of testing scenarios due to the strict Gaussian assumption it relies upon.

**Discussion.** Empirical examination of a toy dataset presented below reveals a case where the GEM's Gaussian assumption may prove inadequate, as demonstrated in Fig 1. For the purpose of visualization, we begin by considering a simple scenario in which the input distribution is a mixture of two-dimensional Gaussians with means $\mu_2 = 2\mu_1 = 8$ and variances $\sigma_1 = \sigma_2 = 1$ respectively. While the GEM can well align with the true data density function, the alignment of GEM scores with the true data density is noticeably compromised when the $p(\mathbf{x})$ is changed to a mixture of Gamma and Gaussian distributions (as shown on the right). In order to ensure an accurate estimation of the ID class-conditional density, two fundamental questions arise:

- ♣ Can we develop a unified framework that guides the design of $g_{\boldsymbol{\theta}}(\mathbf{z}, k)$?
- ♠ Within this framework, how can we obtain a tractable estimate for $\Phi(k)$ without presuming any particular prior distribution of $\hat{p}_{\boldsymbol{\theta}}(\mathbf{z}|k)$?

In the following section, we propose a novel theoretical framework to answer the above questions.

## 3 METHODOLOGY

In this section, we first present the main Bregman Divergence-based theoretical framework of OOD detection in Sec. 3.1. This framework unifies density function formulation and connects with prior OOD techniques, leveraging the expansive exponential distribution family. Motivated by theory, we introduce a novel approach called CONJNORM to determine the desired $g_{\boldsymbol{\theta}}$ through an exhaustive search for the best norm coefficient $p$. To enable tractable density estimation, we explore two partition function estimation baselines and propose our importance sampling in Sec. 3.2.

### 3.1 BREGMAN DIVERGENCE-GUIDED DESIGN OF $g_{\theta}(\mathbf{z}, k)$

In formulating our theoretical framework, it is imperative to adopt a universal distribution family to model the ID class-conditioned distributions $\hat{p}_{\boldsymbol{\theta}}(\mathbf{x}|k)$ without constraining ourselves to any particular choice. In this work, we consider the broad *Exponential Family of Distributions* (Brown, 1986). The family encompasses a wide range of probability distributions frequently employed in prior OOD investigations, such as Gaussian, Gibbs-Boltzmann, and gamma distributions. To be precise, the exponential family of distribution can be formally defined as follows:

**Definition 1** (Exponential Family of Distribution (Brown, 1986)). *A regular exponential family $\hat{p}_{\boldsymbol{\theta}}(\mathbf{z}|k)$ is a family of probability distributions with density function with the parameters $\boldsymbol{\eta}_k$:*

$$\hat{p}_{\boldsymbol{\theta}}(\mathbf{z}|k) = \exp\{\mathbf{z}^{\top}\boldsymbol{\eta}_k - \psi(\boldsymbol{\eta}_k) - g_{\psi}(\mathbf{z})\}, \tag{6}$$

*where $\psi(\cdot)$ is the so-called cumulant function and is a convex function of Legendre type.*

By employing different cumulant functions $\psi(\cdot)$ and parameters $\boldsymbol{\eta}_k$, one can create diverse class-conditioned distributions $\hat{p}_{\boldsymbol{\theta}_k}(\mathbf{z}|k)$. Nevertheless, it has been argued by Azoury & Warmuth (2001); Chowdhury et al. (2023) that directly learning $\boldsymbol{\eta}_k$ to fit the ID data is computationally costly and even *intractable*. To mitigate this challenge, a corresponding dual theorem (referred to as Theorem 1) has been developed. This theorem asserts that any regular exponential family distribution can be presented through a *uniquely* determined *Bregman divergence* (Bregman, 1967), as defined below:

**Definition 2** (Bregman Divergence (Bregman, 1967)). *Let $\varphi(\cdot)$ be a differentiable, strictly convex function of the Legendre type, the Bregman divergence is defined as:*

$$d_{\varphi}(\mathbf{z}, \mathbf{z}') = \varphi(\mathbf{z}) - \varphi(\mathbf{z}') - (\mathbf{z} - \mathbf{z}')^{\top}\nabla\varphi(\mathbf{z}'), \tag{7}$$

*where $\nabla\varphi(\mathbf{z}')$ represents the gradient vector of $\varphi(\cdot)$ evaluated at $\mathbf{z}'$.*

The choices of the convex function $\varphi$ in Bregman divergence can result in diverse distance metrics. For instance, 1) When $\varphi(\mathbf{z}) = \|\mathbf{z}\|^2$, the resulting $d_{\varphi}$ corresponds to the squared Euclidean distance; 2) When $\varphi(\mathbf{z})$ is the negative entropy function, $d_{\varphi}$ represents the KL divergence; and 3) When $\varphi(\mathbf{z})$ be expressed as a quadratic form, $d_{\varphi}$ represents the Mahalanobis distance. Next, Theorem 1 bridges the Bregman divergence and the exponential family of distributions.

**Theorem 1** (Forster & Warmuth (2002)). *Suppose that $\psi(\cdot)$ and $\varphi(\cdot)$ are conjugate Legendre functions. Let $\hat{p}_{\boldsymbol{\theta}}(\mathbf{z}|k)$ be a member of the exponential family conditioned on the $k$-th ID class with cumulant function $\varphi$ and parameters $\boldsymbol{\eta}_k$ $(k = 1, ..., K)$, $d_{\varphi}$ be the Bregman divergence, then $\hat{p}_{\boldsymbol{\theta}}(\mathbf{z}|k)$*

*can be represented as follows: $\hat{p}_{\boldsymbol{\theta}}(\mathbf{z}|k) = \exp(-d_\varphi(\mathbf{z}, \boldsymbol{\mu}(\boldsymbol{\eta}_k)) - g_\varphi(\mathbf{z}))$, where $\boldsymbol{\mu}(\boldsymbol{\eta}_k)$ is the expectation parameter corresponding to $\boldsymbol{\eta}_k$ (Banerjee et al., 2005), $g_\varphi(\cdot)$ is a function uniquely determined by $\varphi(\cdot)$, and agnostic to $\boldsymbol{\mu}(\boldsymbol{\eta}_k)$.*

**Remark.** As a direct implication of Theorem 1, a unified theoretical principle emerges for the design of $g_{\boldsymbol{\theta}}(\mathbf{z}, k)$ for OOD detection, owing to the *conjugate* relationship between $\psi$ and $\varphi$. In essence, when seeking an appropriate $\psi$ for a given dataset, the optimal design of $g_{\boldsymbol{\theta}}(\mathbf{z}, k)$ should inherently adhere to the requirements of the corresponding Bregman divergence: let $\varphi(\cdot) = \psi^*(\cdot)$, then

$$g_{\boldsymbol{\theta}}(\mathbf{z}, k) = \exp(-d_\varphi(\mathbf{z}, \boldsymbol{\mu}(\boldsymbol{\eta}_k))). \tag{8}$$

Given that $g_\varphi(\mathbf{z})$ is agnostic to the choice of $z$, we exclude this term from consideration by treating it as a constant in our analysis, which gives us a systematic approach to answering the question ♣.

**CONJNORM.** Given the expansive function space for the selection of the convex function $\psi$, our focus is on simplifying the search process by utilizing the $l_p$ norm as $\psi$, denoted as CONJNORM, where $\psi(\boldsymbol{\eta}_k) = \frac{1}{2}\|\boldsymbol{\eta}_k\|_p^2$. Therefore, the task of selecting an appropriate $\psi$ is equivalent to identifying a suitable $p$ from the range of $(1, +\infty)$ for the given dataset. The $l_p$ norm offers several advantageous properties, including convexity and simplicity in its conjugate pair. Firstly, the $l_p$ norm is convex for all $p \geq 1$, ensuring the presence of a global minimum during optimization. Secondly, the $l_p$ norm has a well-defined and simple conjugate pair, namely the $l_q$ norm, where $q$ represents the conjugate exponent of $p$ such that $1/p + 1/q = 1$. This simplicity in the conjugate pair enhances computational tractability and facilitates the determination of $\varphi = \psi^*$:

$$\varphi(\mathbf{z}) = \psi^*(\mathbf{z}) = \frac{1}{2}\|\mathbf{z}\|_q^2, \text{ where } q = \frac{p}{p-1}. \tag{9}$$

To this end, the desired Bregman divergence $d_\varphi$ can be determined as

$$d_\varphi(\mathbf{z}, \boldsymbol{\mu}(\boldsymbol{\eta}_k)) = \frac{1}{2}\|\mathbf{z}\|_q^2 + \frac{1}{2}\|\boldsymbol{\mu}(\boldsymbol{\eta}_k)\|_q^2 - \langle \mathbf{z}, \nabla\frac{1}{2}\|\boldsymbol{\mu}(\boldsymbol{\eta}_k)\|_q^2 \rangle. \tag{10}$$

Hence, our final ID density can be estimated by combining Eq. 6 and Theorem 1

$$\hat{p}_{\boldsymbol{\theta}}(\mathbf{z}) = \frac{1}{K}\sum_{k=1}^{K}\frac{g_{\boldsymbol{\theta}}(\mathbf{z}, k)}{\Phi(k)} = \frac{1}{K}\sum_{k=1}^{K}\frac{\exp(-d_\varphi(\mathbf{z}, \boldsymbol{\mu}(\boldsymbol{\eta}_k)))}{\int \exp(-d_\varphi(\mathbf{z}', \boldsymbol{\mu}(\boldsymbol{\eta}_k)))\mathrm{d}\mathbf{z}'}. \tag{11}$$

In the context of our CONJNORM framework, where we treat $p$ as a hyperparameter, the process of searching for the optimal $p^{opt}$ and identifying the most suitable density function $d_\varphi$ for a given dataset becomes straightforward. We present experimental results that explore the effects of varying $p$ as illustrated in Fig. 4.

## 3.2 ESTIMATION OF PARTITION FUNCTION $\Phi(\cdot)$

To an estimate of $\hat{p}_{\boldsymbol{\theta}}$ in Eq. 11, it is imperative to accurately approximate the partition function $\Phi(k)$. The most straightforward approach to address this challenge involves fitting $k$ distinct *kernel density functions*, each corresponding to a different class. By employing this method, the density function $g_{\boldsymbol{\theta}}$ can be effectively normalized:

**Baselines 1: Self-Normalization (SN).** Following Gutmann & Hyvärinen (2012b); Wu et al. (2018); Mnih & Kavukcuoglu (2013), we assume the pre-trained neural network is perfectly expressively such that the unnormalized density function $d_\varphi$ is self-normalized, *i.e.,* $\Phi(k) = \text{constant}, \forall k \in \mathcal{Y}$. In this way, there is no need to explicitly compute the partition function $\Phi(k)$, which is given by

$$\hat{p}_{\boldsymbol{\theta}}(\mathbf{z}) \propto \sum_{k=1}^{K}\exp(-d_\varphi(\mathbf{z}, \boldsymbol{\mu}(\boldsymbol{\eta}_k))). \tag{12}$$

**Baselines 2: Normalization via Kernel Density Estimation.** Kernel density estimation (KDE) (Chen, 2017; Kim & Scott, 2012) is a statistical method that is commonly used for probability density estimation. This approach is inherently non-parametric, providing flexibility in the choice of kernel functions (*e.g.*, linear, Gaussian, exponential). Mathematically, the KDE-based estimation of partition function $\Phi(k)$ can be formulated as follows: let $\mathcal{D}_{\text{in}}^k$ be the ID training data with label $k$,

$$\Phi_{\text{KDE}}(k) = \frac{1}{h|\mathcal{D}_{\text{in}}^k|g_{\boldsymbol{\theta}}(\mathbf{z}, k)}\sum_{\mathbf{z}' \in \mathcal{D}_{\text{in}}^k}\mathcal{K}(\frac{g_{\boldsymbol{\theta}}(\mathbf{z}, k) - g_{\boldsymbol{\theta}}(\mathbf{z}', k)}{h}), \tag{13}$$

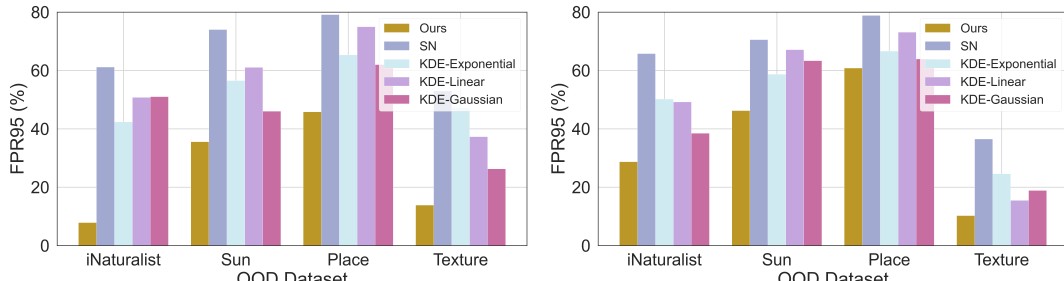

Figure 2: Evaluations of different partition function estimation baselines on ImageNet: Left: MobileNetV2 and Right: ResNet50.

with $h > 0$ as the bandwidth that determines the smoothing of the resulting density function.

Experimental comparisons w.r.t. different baselines on the OOD detection benchmarks are summarized in Fig 2. As we demonstrate later, we propose to leverage the means of importance sampling for theoretically unbiased estimation instead, which provides stronger flexibility and generality.

**Ours: Importance Sampling-Based Approximation.** To enhance the flexibility of density-based OOD detection, we consider a Monte Carlo method and construct a simple and analytically tractable estimator to theoretically unbiasedly approximate them by means of importance sampling (IS) (Liu et al., 2015; Tokdar & Kass, 2010; Ben Alaya et al., 2023). Specifically, let $\hat{p}_o(\mathbf{z})$ be a tractable distribution that has been properly normalized such that $\int \hat{p}_o(\mathbf{z})\,d\mathbf{z} = 1$, we draw data $S = \left\{ (\mathbf{z}_o^1, \mathbf{y}_o^1), ..., (\mathbf{z}_o^n, \mathbf{y}_o^n) \right\}$ from the ID training data following the distribution $\hat{p}_o(\mathbf{z})$, and estimate $\Phi(k)$ by

$$\Phi_{\text{IS}}(k; S) = \frac{1}{n} \sum_{i=1}^{n} \frac{g_{\boldsymbol{\theta}}(\mathbf{z}_o^i, k)}{\hat{p}_o(\mathbf{z}_o^i)}. \tag{14}$$

For simplicity, we set $\hat{p}_o$ as a uniform distribution over the training ID data $\mathcal{D}_{\text{in}}$ and $n = N \times \alpha$ with $\alpha$ as the sampling ratio. In practice, we find that $\alpha = 10\%$ is sufficient to get decent performance. Besides, a desirable property of importance sampling is that the estimator $\Phi_{\text{IS}}(k; S)$ is theoretically unbiased (Liu et al., 2015), *i.e.,* $\mathbb{E}_{S \sim \hat{p}_o}[\Phi_{\text{IS}}(k; S)] = \Phi(k)$. IS provides us with a simple but effective approach to answering the question ♠.

## 4 EXPERIMENTS

### 4.1 EXPERIMENTS SETUP

**Baseline Methods**. We compare our method with representative methods, including MSP (Hendrycks & Gimpel, 2016), ODIN (Liang et al., 2017), Energy (Liu et al., 2020), ASH (Djurisic et al., 2022), DICE (Sun & Li, 2022), ReAct (Sun et al., 2021), Mahalanobis (Maha) (Lee et al., 2018), GEM (Morteza & Li, 2022), KNN (Sun et al., 2022) and SHE (Zhang et al., 2022). It is worth noting that we have adopted the recommended configurations proposed by prior works, while concurrently standardizing the backbone architecture to ensure equitable comparisons.

**Evaluation Metrics**. The detection performance is evaluated via two threshold-independent metrics: the false positive rate of OOD data is measured when the true positive rate of ID data reaches 95% (FPR95); and the

Table 1: OOD detection on CIFAR benchmarks. We average the results across 6 OOD datasets. ↑ indicates larger values are better and vice versa. The best result in each column is shown in bold.

| Method | CIFAR-10 | | CIFAR-100 | |
|---|---|---|---|---|
| | FPR95↓ | AUROC↑ | FPR95↓ | AUROC↑ |
| MSP | 48.73 | 92.46 | 80.13 | 74.36 |
| ODIN | 24.57 | 93.71 | 58.14 | 84.49 |
| Energy | 26.55 | 94.57 | 68.45 | 81.19 |
| DICE | 20.83 | 95.24 | 49.72 | 87.23 |
| ReAct | 26.45 | 94.67 | 62.27 | 84.47 |
| ASH | 15.05 | 96.61 | 41.40 | 90.02 |
| Maha | 31.42 | 89.15 | 55.37 | 82.73 |
| GEM | 29.56 | 92.14 | 49.31 | 86.45 |
| KNN | 17.43 | 96.74 | 41.52 | 88.74 |
| SHE | 23.26 | 94.40 | 54.66 | 82.60 |
| Ours | 13.92 | 97.15 | 28.27 | 92.50 |
| Ours+ASH | **12.14** | **97.64** | **25.66** | **93.21** |

area under the receiver operating characteristic curve (AUROC) is computed to quantify the probability of the ID case receiving a higher score compared to the OOD case. Reported performance

Table 2: OOD detection results on the ImageNet benchmark with MobileNet-V2. ↑ indicates larger values are better and vice versa. The best result in each column is shown in bold.

| Method | iNaturalist | | SUN | | Places | | Textures | | Average | |
|---|---|---|---|---|---|---|---|---|---|---|
| | FPR95↓ | AUROC↑ | FPR95↓ | AUROC↑ | FPR95↓ | AUROC↑ | FPR95↓ | AUROC↑ | FPR95↓ | AUROC↑ |
| MSP | 64.29 | 85.32 | 77.02 | 77.10 | 79.23 | 76.27 | 73.51 | 77.30 | 73.51 | 79.00 |
| ODIN | 55.39 | 87.62 | 54.07 | 85.88 | 57.36 | 84.71 | 49.96 | 85.03 | 54.20 | 85.81 |
| Energy | 59.50 | 88.91 | 62.65 | 84.50 | 69.37 | 81.19 | 58.05 | 85.03 | 62.39 | 84.91 |
| ReAct | 42.40 | 91.53 | 47.69 | 88.16 | 51.56 | 86.64 | 38.42 | 91.53 | 45.02 | 89.60 |
| DICE | 43.09 | 90.83 | 38.69 | 90.46 | 53.11 | 85.81 | 32.80 | 91.30 | 41.92 | 89.60 |
| ASH | 39.10 | 91.94 | 43.62 | 90.02 | 58.84 | 84.73 | 13.12 | 97.10 | 38.67 | 90.95 |
| Maha | 62.11 | 81.00 | 47.82 | 86.33 | 52.09 | 83.63 | 92.38 | 33.06 | 63.60 | 71.01 |
| GEM | 65.77 | 79.82 | 45.53 | 87.45 | 82.85 | 68.31 | 43.49 | 86.22 | 59.39 | 80.25 |
| KNN | 46.78 | 85.96 | 40.18 | 86.28 | 62.46 | 82.96 | 31.79 | 90.82 | 45.30 | 86.51 |
| SHE | 47.61 | 89.24 | 42.38 | 89.22 | 56.62 | 83.79 | 29.33 | 92.98 | 43.98 | 88.81 |
| Ours | 29.06 | 93.89 | 46.74 | 87.10 | 62.07 | 81.41 | **10.30** | **97.53** | 37.04 | 89.98 |
| Ours+ASH | **24.08** | **94.36** | **30.19** | **92.63** | **46.26** | **87.57** | 12.70 | 97.20 | **28.31** | **92.94** |

results for our method are averaged over 5 independent runs for robustness. Due to the space limit, we provide the implementation details in the Appendix.

## 4.2 MAIN RESULTS

**Evaluation on CIFAR Benchmarks.** Following the setup in Sun & Li (2022), we consider CIFAR-10 and CIFAR-100 (Krizhevsky et al., 2009) as ID data and train DenseNet-101 (Huang et al., 2017) on them respectively using the cross-entropy loss. The feature dimension of the penultimate layer is 342. For both CIFAR-10 and CIFAR-100, the model is trained for 100 epochs, with batch size 64, weight decay 1e-4, and Nesterov momentum 0.9. The start learning rate is 0.1 and decays by a factor of 10 at 50th, 75th, and 90th epochs. There are six datasets for OOD detection with regard to CIFAR benchmarks: SVHN (Netzer et al., 2011), LSUN-Crop (Yu et al., 2015), LSUN-Resize (Yu et al., 2015), iSUN (Xu et al., 2015), Places (Zhou et al., 2017), and Textures (Cimpoi et al., 2014). At test time, all images are of size 32×32. Table 1 presented the performance of our approach and existing competitive baselines, where the proposed approach significantly outperforms existing methods. Specifically, comparing with the standard post-hoc methods, our method reveals 3.51% and 0.41% average improvements w.r.t. FPR95 and AUROC on the CIFAR-10 dataset, and 13.25% and 3.76% of the average improvements on the CIFAR-100 dataset. For advanced works that consider post-hoc enhancement, *e.g.,* ASH and DICE, our method still significantly performs better on both datasets

**Evaluation on ImageNet Benchmark.** We conduct experiments on the ImageNet benchmark, demonstrating the scalability of our method. Specifically, we inherit the exact setup from (Djurisic et al., 2022), where the ID dataset is ImageNet-1k (Krizhevsky et al., 2012), and OOD datasets include iNaturalist (Xiao et al., 2010b), SUN (Xiao et al., 2010a), Places365 (Zhou et al., 2017), and Textures (Cimpoi et al., 2014). We use the pre-trained MobileNetV2 (Sandler et al., 2018) models for ImageNet-1k provided by Pytorch (Paszke et al., 2019). At test time, all images are resized to 224×224. In Table 2, we reported the performances of four OOD test datasets respectively. It can be seen that our method reaches state-of-the-art with 21.51% FPR95 and 95.48% AUROC on average across four OOD datasets. Besides, we notice that ASH can further considerably improve our method by 9.27% and 2.06% w.r.t. FPR95 and AUROC respectively. We suspect that removing a large portion of activations at a late layer helps to improve the representative ability of features.

## 4.3 ABLATION STUDY

**Extracted Features z.** This paper follows the convention in feature-based OOD detectors (Sun et al., 2022; Zhang et al., 2022; Djurisic et al., 2022), where features from the penultimate layer are utilized to estimate uncertain scores for OOD detection. Fig 3 provides an experimental evaluation on the choice of working placement. It can be seen that feature from the deeper layer contributes to better OOD detection performance than shallower ones. This is likely due to the penultimate layer preserves more information than shallower layers.

**Sampling Ratio α.** In Fig 4, we analyze the effect of the sampling ratio $\alpha$ on CIFAR-100 and ImageNet-1k datasets. We vary the random sampling ratio $\alpha$ within $\{1\%, 5\%, 10\%, 50\%, 100\%\}$. We note several interesting observations: (1) The optimal OOD detection (measured by FPR95) remains similar under different random sampling ratios $\alpha$ especially when $\alpha \geq 10\%$, which demon-

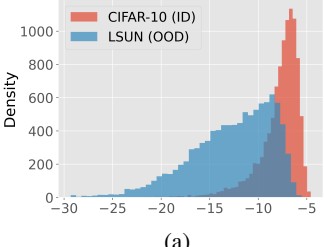 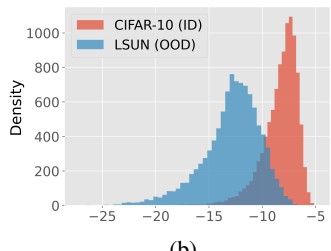 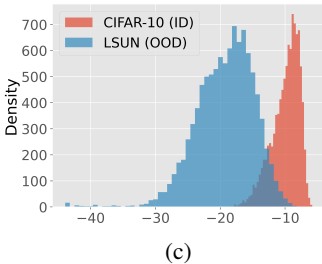

(a)                (b)                (c)

Figure 3: Ablation study using feature extractions from (a) the first, (b) the second, and (c) the last dense block of the DenseNet on the CIFAR-10.

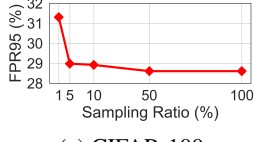 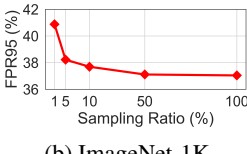 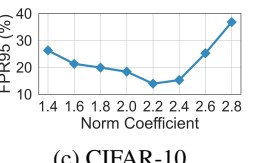 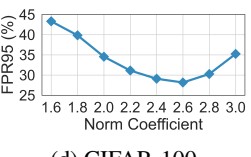

(a) CIFAR-100       (b) ImageNet-1K       (c) CIFAR-10       (d) CIFAR-100

Figure 4: Ablation study w.r.t varing sampling ratio $\alpha$ in red; and the norm coefficient $p$ in blue.

strates the robustness of our method to the sampling ratio. (2) our method still achieves competitive performance on benchmarks even when sampling $1\%$ total number of ID training data.

**Parameter Sensitivity of $l_p$.** We conduct a comparative assessment of OOD performance while varying the $l_p$ norm coefficient $p$, which directly governs the density function $g_{\boldsymbol{\theta}}$ on CIFAR benchmarks. The results, as depicted in Fig 4, reveal a consistent trend across both datasets. Notably, the FPR95 scores exhibit a clear minimum within the range of $(2, 3)$, suggesting that our proposed CONJNORM approach can efficiently identify the optimal normalization without the computational overhead. It is worth highlighting that when $p = 2$, signifying the Bregman divergence in Eq.(10) degenerates into the squared Euclidean distance (corresponding to Gaussian densities), the OOD performance does not attain its peak. This observation underscores the limitation of Gaussian assumptions and underscores the generality and effectiveness of our CONJNORM.

**Sensitivity of $q$ with fixed $l_p$ norm.** We also conduct a comparative assessment of OOD performance by varying the value of $q$ while fixing the $l_p$ norm. The experimental results on CIFAR-100 under two cases where $p = 2.5$ and $p = 3.0$. Note that the performance of our method tends to be more appealing when $q$ satisfies the conjugate condition that $q = p/(p - 1)$, exceeding the case where $q = 2.0$ by nearly $10\%$ on FPR95. This empirically echoes Theorem 1.

### 4.4 EXTENSION TO MORE PROTOCOLS

In this section, we assess the versatility of the proposed CONJNORM approach in (1) Hard OOD detection and (2) Long-tailed OOD settings. For more extensions, please refer to the Appendix.

#### 4.4.1 HARD OOD DETECTION

We consider hard OOD scenarios (Tack et al., 2020), of which the OOD data are semantically similar to that of the ID cases. With the CIFAR-100 as the ID dataset for training ResNet-50. we evaluate our

Table 3: Evaluation on hard OOD detection tasks. ↑ indicates larger values are better and vice versa. The best result in each column is shown in bold.

| Method | LSUN-Fix | | ImageNet-Fix | | ImageNet-Resize | | CIFAR-10 | | Average | |
|---|---|---|---|---|---|---|---|---|---|---|
| | FPR95↓ | AUROC↑ | FPR95↓ | AUROC↑ | FPR95↓ | AUROC↑ | FPR95↓ | AUROC↑ | FPR95↓ | AUROC↑ |
| MSP | 90.43 | 63.97 | 88.46 | 67.32 | 86.38 | 71.24 | 89.67 | 66.47 | 88.73 | 67.25 |
| ODIN | 91.28 | 66.53 | 82.98 | 72.89 | 72.71 | 82.19 | 88.27 | 71.30 | 83.81 | 73.23 |
| Energy | 91.35 | 66.52 | 83.02 | 72.88 | 72.45 | 82.22 | 88.17 | 71.29 | 83.75 | 73.23 |
| ReAct | 93.70 | 64.52 | 83.36 | 73.47 | **62.85** | 85.79 | 89.09 | 69.87 | 82.25 | 73.41 |
| Maha | 90.54 | 56.43 | 83.24 | 61.84 | 75.83 | 64.71 | 90.27 | 54.36 | 84.97 | 59.33 |
| KNN | 91.70 | 69.70 | 80.58 | 76.46 | 68.90 | 85.98 | **83.28** | 75.57 | 81.12 | 76.93 |
| SHE | 93.52 | 63.56 | 85.62 | 70.75 | 81.54 | 76.97 | 89.32 | 71.52 | 87.50 | 70.70 |
| Ours | **85.80** | **72.48** | **76.14** | **78.77** | 65.38 | **86.29** | 84.87 | **75.88** | **78.05** | **78.35** |

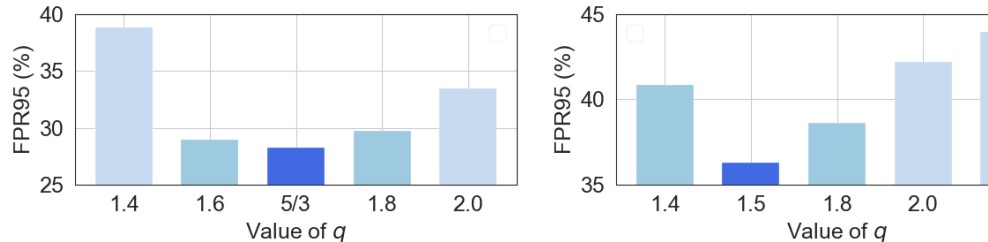

Figure 5: Comparisons of varying $q$ when $p$ is fixed at 2.5 (Left) and 3.0 (Right) on CIFAR-100.

Table 4: Evaluation on long-tailed OOD detection tasks. ↑ indicates larger values are better and vice versa. The best result in each column is shown in bold.

| Method | SVHN | | LSUN | | iSUN | | Texture | | Places365 | | Average | |
|---|---|---|---|---|---|---|---|---|---|---|---|---|
| | FPR95↓ | AUROC↑ | FPR95↓ | AUROC↑ | FPR95↓ | AUROC↑ | FPR95↓ | AUROC↑ | FPR95↓ | AUROC↑ | FPR95↓ | AUROC↑ |
| MSP | 97.82 | 56.45 | 82.48 | 73.54 | 97.61 | 54.95 | 95.51 | 54.53 | 92.49 | 60.08 | 93.18 | 59.91 |
| ODIN | 98.70 | 48.32 | 64.80 | 83.70 | 97.47 | 52.41 | 95.99 | 49.27 | 91.56 | 58.49 | 89.70 | 58.44 |
| Energy | 98.81 | 43.10 | 47.03 | 89.41 | 97.37 | 50.77 | 95.82 | 46.25 | 91.73 | 57.09 | 86.15 | 57.32 |
| Maha | 74.01 | 83.67 | 77.88 | 72.63 | 49.42 | 85.52 | 63.10 | 81.28 | 94.37 | 53.00 | 71.76 | 75.22 |
| MSP+RP | 97.76 | 56.45 | 82.33 | 73.54 | 97.51 | 54.76 | 95.73 | 54.13 | 92.65 | 59.73 | 93.20 | 59.72 |
| ODIN+RW | 98.82 | 52.94 | 83.79 | 69.28 | 96.10 | 50.64 | 96.95 | 45.14 | 93.36 | 51.5 | 93.80 | 53.90 |
| Energy+RW | 98.86 | 49.07 | 77.30 | 77.32 | 96.25 | 50.91 | 96.86 | 45.27 | 93.03 | 54.02 | 92.46 | 55.32 |
| KNN | 64.39 | 86.16 | 56.13 | 84.24 | 45.36 | 88.39 | **34.36** | **89.86** | **90.31** | **60.09** | 58.11 | 81.75 |
| Ours | **40.16** | **91.00** | **45.72** | **87.64** | **41.89** | **90.42** | 40.50 | 86.80 | 91.74 | 58.44 | **52.00** | **82.86** |

method on 4 hard OOD datasets, namely, LSUN-Fix (Yu et al., 2015), ImageNet-Fix (Krizhevsky et al., 2012), ImageNet-Resize (Krizhevsky et al., 2012), and CIFAR-10. The model is trained for 200 epochs, with batch size 128, weight decay 5e-4 and Nesterov momentum 0.9. The start learning rate is 0.1 and decays by a factor of 5 at 60th, 12th, 160th epochs. We select a set of strong baselines that are competent in hard OOD detection, and the experiments are summarized in Table 3. It can be seen that our method can beat the state-of-the-art across the considered datasets, even for the challenging CIFAR-100 versus CIFAR-10 setting. The reason is that our $l_p$ norm-induced density function can better capture the ID data distribution.

### 4.4.2 Long-tailed OOD Detection

We consider long-tailed OOD scenarios (Wang et al., 2022; Bai et al., 2023), of which the ID training data exhibits an imbalanced class distribution. We use the long-tailed versions of CIFAR datasets with the setting in Cao et al. (2019); Zhong et al. (2021). It is by controlling the degrees of data imbalance with an imbalanced factor $\beta = N_{max}/N_{min}$, where $N_{max}$ and $N_{min}$ are the numbers of training samples belonging to the most and the least frequent classes. Following (Zhong et al., 2021; Zhou et al., 2020), we pre-train the ResNet-32 (He et al., 2016) network with $\beta = 50$ on CIFAR-100 for 200 epochs with batch size 128, weight decay 2e-4 and Nesterov momentum 0.9. The start learning rate is 0.1 and decays by a factor of 5 at the 160-th, 180-th epochs. The performance of our methods and baselines are shown in Table 4, where we introduce the strategies of Replacing (RP) and Reweighting (RW) in (Jiang et al., 2023) to modify previous OOD scoring functions. The performance gain in Table 4 empirically demonstrates that using a uniform ID class distribution does not make our method incompatible with the model that is pre-trained with class-imbalanced data.

## 5 Conclusion

In this paper, we present a theoretical framework for studying density-based OOD detection. By establishing connections between the exponential family of distributions and Bregman divergence, we provide a unified principle for designing scoring functions. Given the expansive function space for selecting Bregman divergence, we propose a pair of conjugate functions to simplify the search process. To address the challenging problem of the partition function, we introduce a computationally tractable and theoretically unbiased estimator through importance sampling. Empirically, our method outperforms numerous prior methods by a significant margin on several standard benchmark datasets using various protocols. Since we only consider a pair of conjugate functions in finding Bregman divergence and evaluate our method on Convolutional neural networks. In the future, it is interesting to delve deeper into the design of Bregman divergence and incorporate large-scale pre-trained Vision-Language Models (VLMs).

## 6 ACKNOWLEDGEMENT

This work is partially supported by the Australian Research Council (CE200100025 and DE240100105). Li gratefully acknowledges the support of the AFOSR Young Investigator Program under award number FA9550-23-1-0184, National Science Foundation (NSF) Award No. IIS-2237037 & IIS-2331669, and Office of Naval Research under grant number N00014-23-1-2643.

## 7 ETHIC STATEMENT

This paper does not raise any ethical concerns. This study does not involve any human subjects practices to data set releases, potentially harmful insights, methodologies and applications, potential conflicts of interest and sponsorship, discrimination/bias/fairness concerns, privacy and security issues.legal compliance, and research integrity issues.

## 8 REPRODUCIBILITY STATEMENT

To make all experiments reproducible, we have listed all detailed hyper-parameters. We upload source codes and instructions in the supplementary materials.

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

Table 5: OOD detection on CIFAR100 benchmarks. We average the results across 6 OOD datasets. ↑ indicates larger values are better and vice versa. The best result in each column is shown in bold.

| Metrics | MSP | ODIN | Energy | ReAct | DICE | Maha | GEM | KNN | ASH | Ours |
|---|---|---|---|---|---|---|---|---|---|---|
| FPR95↓ | 79.29 | 77.47 | 76.66 | 69.95 | 73.06 | 71.75 | 65.21 | 62.22 | 65.29 | **54.47** |
| AUROC↑ | 79.9 | 74.36 | 80.14 | 81.99 | 79.86 | 63.14 | 77.81 | 84.22 | 81.36 | **87.26** |

# A APPENDIX

## A.1 LIMITATIONS

The limitation of this work lies in manually searching a good value of $p$ to determine Bregman divergence. We do not test our method on large-scale models.

## A.2 OOD DATASET

For experiments where CIFAR benchmarks are the ID data, we adopt SVHN (Netzer et al., 2011), LSUN-Crop (Yu et al., 2015), LSUN-Resize (Yu et al., 2015), iSUN (Xu et al., 2015), Places (Zhou et al., 2017), and Textures (Cimpoi et al., 2014) as the OOD datasets. For experiments where ImageNet-1K is the ID data, we adopt iNaturalist (Xiao et al., 2010b), SUN (Xiao et al., 2010a), Places365 (Zhou et al., 2017), and Textures (Cimpoi et al., 2014) and the OOD dataset.

## A.3 IMPLEMENTATION DETAILS.

Similar to DICE (Sun & Li, 2022), we adopt Tiny-ImageNet-200 (Le & Yang, 2015) as the auxiliary OOD data with the searching space of $p$ as (1,3]. We remove those data whose labels coincide with ID cases. We set $p = 2.2$ for experiments in CIFAR-10, $p = 2.5$ for experiments in CIFAR-100, $p = 1.5$ and $p = 1.8$ for experiments in ImageNet-1k on ResNet50 and MobileNetv2 respectively.

## A.4 RESULTS WITH DIFFERENT BACKBONES

In the main paper, we have shown that our method is competitive on DenseNet and MobileNet. In this section, we show in Table 5 and Table 6 that the strong performance of our method holds on ResNet50 (He et al., 2016). All the numbers reported are averaged over OOD test datasets described in Section 4.2. For ImageNet-1k, We use the pre-trained models provided by Pytorch. At test time, all images are resized to 224×224. For CIFAR-100, the model is trained for 200 epochs, with batch size 128, weight decay 5e-4 and Nesterov momentum 0.9. The start learning rate is 0.1 and decays by a factor of 5 at 60th, 120th and 160th epochs. At test time, all images are of size 32×32.

## A.5 OOD DETECTION WITH CONTRASTIVE REPRESENTATIONS

We explore the compatibility of our method with contrastive representations. Closely following the training protocol in Ming et al. (2022b); Khosla et al. (2020), we pre-train ResNet-34 (He et al., 2016) on CIFAR-100 with the SupCon (Khosla et al., 2020) and CIDER (Ming et al., 2022b) losses respectively. We train the model using stochastic gradient descent for 500 epochs with batch size 512, Nesterov momentum 0.9, and weight decay 1e-4. The initial learning rate is 0.5 with cosine scheduling. Table 7 demonstrates, under both SupCon and CIDER settings, ours consistently outperforms the Maha, SHE and KNN scores by a large margin, highlighting our method's effectiveness.

## A.6 MORE RESULTS ON LONG-TAILED OOD DETECTION

Continuing from Section 4.4.2, we further test the compatibility of our method to the model that is pre-trained with class-imbalanced data. In this section, we use the long-tailed versions of CIFAR-10 datasets with the setting in Cao et al. (2019); Zhong et al. (2021) with an imbalanced factor $\beta = 50$. Following (Zhong et al., 2021; Zhou et al., 2020), we pre-train the ResNet-32 (He et al., 2016) network with $\beta = 50$ on CIFAR-10 for 200 epochs with batch size 128, weight decay 2e-4 and Nesterov momentum 0.9. The start learning rate is 0.1 and decays by a factor of 5 at the 160-th, 180-th epochs. The performance of our methods and baselines are shown in Table 8.

Table 6: OOD detection results on the ImageNet benchmark with ResNet-50. ↑ indicates larger values are better and vice versa. The best result in each column is shown in bold.

| Method | iNaturalist | | SUN | | Places | | Textures | | Average | |
|---|---|---|---|---|---|---|---|---|---|---|
| | FPR95↓ | AUROC↑ | FPR95↓ | AUROC↑ | FPR95↓ | AUROC↑ | FPR95↓ | AUROC↑ | FPR95↓ | AUROC↑ |
| MSP | 54.99 | 87.74 | 70.83 | 80.86 | 73.99 | 79.76 | 68.00 | 79.61 | 66.95 | 81.99 |
| ODIN | 47.66 | 89.66 | 60.15 | 84.59 | 67.89 | 81.78 | 50.23 | 85.62 | 56.48 | 85.41 |
| Energy | 55.72 | 89.95 | 59.26 | 85.89 | 64.92 | 82.86 | 53.72 | 85.99 | 58.41 | 86.17 |
| ReAct | 20.38 | 96.22 | **24.20** | **94.20** | **33.85** | **91.58** | 47.30 | 89.80 | 31.43 | 92.95 |
| DICE | 25.63 | 94.49 | 35.15 | 90.83 | 46.49 | 87.48 | 31.72 | 90.30 | 34.75 | 90.77 |
| Maha | 97.00 | 52.65 | 98.50 | 42.41 | 98.40 | 41.79 | 55.80 | 85.01 | 87.43 | 55.47 |
| GEM | 51.67 | 81.66 | 68.87 | 73.78 | 79.52 | 67.34 | 35.73 | 86.54 | 58.95 | 77.33 |
| KNN | 59.77 | 85.89 | 68.88 | 80.08 | 78.15 | 74.10 | 10.90 | 97.42 | 54.68 | 84.37 |
| SHE | 45.35 | 90.15 | 45.09 | 87.93 | 54.19 | 84.69 | 34.22 | 90.18 | 44.71 | 88.24 |
| Ours | **9.62** | **97.97** | 37.75 | 90.13 | 48.99 | 86.60 | **9.61** | **97.74** | **26.49** | **93.11** |

Table 7: OOD detection results on CIFAR-100 under contrastive learning. ↑ indicates larger values are better and vice versa. The best result in each column is shown in bold.

| Method | SVHN | | LSUN | | iSUN | | Texture | | Places365 | | Average | |
|---|---|---|---|---|---|---|---|---|---|---|---|---|
| | FPR95↓ | AUROC↑ | FPR95↓ | AUROC↑ | FPR95↓ | AUROC↑ | FPR95↓ | AUROC↑ | FPR95↓ | AUROC↑ | FPR95↓ | AUROC↑ |
| SupCon+Maha | **14.47** | **97.31** | 92.81 | 67.81 | 79.33 | 80.71 | 50.35 | 79.79 | 95.93 | 54.17 | 66.58 | 75.96 |
| SupCon+SHE | 28.35 | 94.34 | 72.53 | 78.81 | 99.18 | 19.31 | 72.16 | 68.07 | 83.17 | 75.41 | 71.08 | 67.19 |
| SupCon+KNN | 38.68 | 92.61 | 43.16 | 91.43 | 67.89 | 85.04 | 58.46 | 86.65 | 75.18 | 77.22 | 56.67 | 86.59 |
| SupCon+Ours | 28.74 | 94.58 | **15.43** | **97.28** | **56.97** | **88.54** | **46.03** | **90.11** | **74.97** | **77.65** | **44.43** | **89.63** |
| CIDER+Maha | 18.52 | 96.36 | 88.86 | 71.93 | 79.55 | 79.76 | 54.31 | 77.89 | 95.47 | 52.93 | 67.34 | 75.77 |
| CIDER+SHE | 28.16 | 94.09 | 69.06 | 82.60 | 99.69 | 15.51 | 75.57 | 62.84 | 87.21 | 72.41 | 71.94 | 65.51 |
| CIDER+KNN | 22.93 | 95.17 | 16.17 | 96.33 | 71.62 | 80.85 | 45.35 | 90.08 | 74.12 | 67.25 | 46.23 | 87.31 |
| CIDER+Ours | **15.87** | **96.55** | **6.04** | **98.76** | **52.45** | **88.32** | **31.97** | **93.31** | **72.31** | **75.94** | **35.73** | **90.58** |

## A.7 DETAILED CIFAR RESULTS

Table 10 and Table 11 supplement Table 1 in the main text, as they display the full results on each of the 6 OOD datasets for DenseNet trained on CIFAR-10 and CIFAR-100 respectively. Table 12 supplement Table 5, as it displays the full results on each of the 6 OOD datasets for ResNet50 trained on CIFAR-100 respectively

## A.8 DISCUSSION ON DEEP GENERATIVE MODELS FOR DENSITY ESTIMATION

Since density estimation plays a key role in our method, our work is related to deep generative models that achieve empirically promising results based on neural networks. Generally, there are two families of DGMs for density estimation: 1) autoregressive models (Germain et al., 2015; Uria et al., 2016; Papamakarios et al., 2017) that decompose the density into the product of conditional densities based on probability chain rule where Each conditional probability is modeled by a parametric density (e.g., Gaussian or mixture of Gaussian) whose parameters are learned by neural networks, and 2) normalizing flows (Rezende & Mohamed, 2015; Ballé et al., 2015; Dinh et al., 2016; Grover et al., 2018; Albergo & Vanden-Eijnden, 2022) that represent input as an invertible transformation of a latent variable with known density with the invertible transformation as a composition of a series of simple functions. While using DGMs for density estimation seems to be a valid and intuitive option for density-based OOD detection, this requires training a DGM from scratch and therefore violates the principle of post-hoc OOD detection, i.e., only pre-trained models at hand are expected to be used to detect OOD data from streaming data at the interference stage. Besides, Zhang et al. (2021) finds that DGMS tend to assign higher probabilities or densities to OOD images than images from the training distribution. We also explore the possibility of integrating pre-trained Diffusion models (Peebles & Xie, 2023; Rombach et al., 2022) into zero-shot class-conditioned density estimation based on Eq.(1) in Li et al. (2023). Unfortunately, the computation is intractable due to the integral. Although authors in Li et al. (2023) use a simplified ELBO for approximation, there is no theoretical guarantee that the ELBO can well align with the data density not to mention the computational-inefficient inference of diffusion models. We will leave this challenge as our future work.

Table 8: Evaluation on long-tailed OOD detection tasks on CIFAR-10. $\uparrow$ indicates larger values are better and vice versa. The best result in each column is shown in bold.

| Method | SVHN | | LSUN | | iSUN | | Texture | | Places365 | | Average | |
|---|---|---|---|---|---|---|---|---|---|---|---|---|
| | FPR95$\downarrow$ | AUROC$\uparrow$ | FPR95$\downarrow$ | AUROC$\uparrow$ | FPR95$\downarrow$ | AUROC$\uparrow$ | FPR95$\downarrow$ | AUROC$\uparrow$ | FPR95$\downarrow$ | AUROC$\uparrow$ | FPR95$\downarrow$ | AUROC$\uparrow$ |
| MSP | 90.12 | 78.00 | 83.62 | 86.98 | 53.29 | 90.00 | 77.23 | 77.24 | 87.42 | 77.88 | 78.34 | 82.02 |
| ODIN | 99.43 | 48.10 | 78.52 | 79.10 | 42.64 | 88.54 | 77.22 | 74.59 | 84.40 | 63.45 | 76.44 | 70.76 |
| Energy | 99.74 | 35.80 | 75.83 | 77.12 | **42.15** | 88.29 | 79.13 | 71.71 | 83.78 | 62.56 | 76.13 | 67.10 |
| Maha | 80.80 | 81.86 | 98.07 | 67.41 | 75.34 | 85.17 | 74.66 | 80.83 | 92.51 | 68.22 | 84.28 | 76.70 |
| MSP+RP | 82.64 | 78.00 | 67.21 | 86.98 | 50.30 | 90.00 | 75.80 | 77.24 | 80.71 | 77.88 | 71.33 | 82.02 |
| ODIN+RW | 88.64 | 70.74 | **52.37** | 84.93 | 63.97 | 84.37 | 82.62 | 68.93 | 74.70 | **79.47** | 72.46 | 77.69 |
| Energy+RW | 94.10 | 67.02 | 54.60 | 87.21 | 52.19 | 89.10 | 79.65 | 71.87 | 75.90 | 80.42 | 71.29 | 79.12 |
| KNN | 77.29 | 86.80 | 77.90 | 78.91 | 50.98 | **93.54** | 76.56 | **88.73** | **57.70** | 77.31 | 68.09 | 85.06 |
| Ours | **28.89** | **93.65** | 55.36 | **85.55** | 48.08 | 88.78 | **54.45** | 85.29 | 78.77 | 79.05 | **53.11** | **86.46** |

## A.9 INTRACTABLE LEARNING OF THE EXPONENTIAL FAMILY NATURAL PARAMETER

Given the fact that $\int \hat{p}_{\boldsymbol{\theta}}(\mathbf{z}|k)\,\mathrm{d}\mathbf{z} = 1$, we then have:

$$\int \exp\left\{\mathbf{z}^\top \boldsymbol{\eta}_k - \psi(\boldsymbol{\eta}_k) - g_\psi(\mathbf{z})\right\}\mathrm{d}\mathbf{z} = 1 \tag{15}$$

Eq. 15 means that, for any known $\psi(\cdot)$ and $g\_\psi(\cdot)$, one can learn the natural parameter $\boldsymbol{\eta}_k$ by solving the following:

$$\exp\psi(\boldsymbol{\eta}_k) = \int \exp\left\{\mathbf{z}^\top \boldsymbol{\eta}_k - g_\psi(\mathbf{z})\right\}\mathrm{d}\mathbf{z} \tag{16}$$

Since the right side of Eq. 16 includes the integral over latent feature space that is high-dimensional, learning the natural parameter of an Exp. Family is said to be intractable.

## A.10 THEORETICAL JUSTIFICATION

Let $\mathcal{B}$ denotes the Borel $\sigma$-algebra on $\mathcal{Z}$ and $\mathcal{P}(\mathcal{Z})$ denotes the set of all probability measures on $(\mathcal{Z}, \mathcal{B})$, We recall the following definitions:

**Definition 3** (Total Variation). *Let $\mathcal{P}_1, \mathcal{P}_2 \in \mathcal{P}(\mathcal{Z})$. The total variation(TV) is defined by:*

$$\delta(\mathcal{P}_1, \mathcal{P}_2) = \sup_{A \in \mathcal{B}} |\mathcal{P}_1(A) - \mathcal{P}_2(A)| \tag{17}$$

We use the following characterization of TV (See Müller (1997) Theorem 5.4):

**Lemma 1.** *Let $\mathcal{P}_1, \mathcal{P}_2 \in \mathcal{P}(\mathcal{Z})$ and let $\mathcal{F}$ denotes the unit ball in $L^\infty(\mathcal{Z})$, i.e.,*

$$\mathcal{F} := \{f \in L^\infty(\mathcal{Z})|\, \|f\|_\infty \le 1\} \tag{18}$$

*then we have the following characterization for the TV distance,*

$$\delta(\mathcal{P}_1, \mathcal{P}_2) = \sup_{f \in \mathcal{F}} |\mathbb{E}_{\mathbf{z} \in \mathcal{P}_1} f(\mathbf{z}) - \mathbb{E}_{\mathbf{z} \in \mathcal{P}_2} f(\mathbf{z})| \tag{19}$$

Next, let us recall the definition of Kullback–Leibler(KL) divergence,

**Definition 4** (KL Divergence). *Let $\mathcal{P}_1, \mathcal{P}_2 \in \mathcal{P}(\mathcal{Z})$ be two probability measures with density functions $p_1$ and $p_2$ respectively. The KL divergence is defined by*

$$KL(\mathcal{P}_1 \| \mathcal{P}_2) := \int_{\mathbf{z} \in \mathcal{Z}} p_1(\mathbf{z}) \ln \frac{p_1(\mathbf{z})}{p_2(\mathbf{z})}\mathrm{d}\mathbf{z} \tag{20}$$

*whenever the above integral is defined.*

Next, recall the following standard lemma that computes KL divergence between exponential family distributions.

**Lemma 2** (Relation between Bregman Divergences and KL Divergence (Banerjee et al., 2005)). *Let $\mathcal{P}_1, \mathcal{P}_2 \in \mathcal{P}(\mathcal{Z})$ conform to exponential family distributions with the corresponding density functions $p_1$ and $p_2$ are parameterized by $\boldsymbol{\eta}_1$ and $\boldsymbol{\eta}_2$ respectively, then we have the following:*

$$KL(\mathcal{P}_1 \| \mathcal{P}_2) = d_\varphi(\boldsymbol{\mu}(\boldsymbol{\eta}_1), \boldsymbol{\mu}(\boldsymbol{\eta}_2)) \tag{21}$$

*where $d_\varphi(\cdot, \cdot)$ is the so-called Bregman Divergence.*

Next, we recall the following inequality that bounds the TV by KL divergence. (see Tsybakov (2008), Lemma 2.5 and Lemma 2.6])

**Lemma 3** (Pinsker inequality). *Let $\mathcal{P}_1, \mathcal{P}_2 \in \mathcal{P}(\mathcal{Z})$ then we have the following:*

$$\delta(\mathcal{P}_1, \mathcal{P}_2) \leq \sqrt{\frac{1}{2} KL(\mathcal{P}_1 \| \mathcal{P}_2)} \tag{22}$$

**Theorem 2.** *We have the following bound:*

$$\mathbb{E}_{\mathbf{z} \in \mathcal{P}_{in}}[\hat{p}_{\boldsymbol{\theta}}(\mathbf{z})] - \mathbb{E}_{\mathbf{z} \in \mathcal{P}_{out}}[\hat{p}_{\boldsymbol{\theta}}(\mathbf{z})] \leq \alpha \tag{23}$$

*where $\alpha := \frac{1}{K} \sum_{k=1}^{K} \sqrt{\frac{1}{2} d_\varphi(\boldsymbol{\mu}(\boldsymbol{\eta}_k), \boldsymbol{\mu}(\boldsymbol{\eta}_{out}))}$*

Theorem 2 bounds the measure $D$ in terms of Bregman divergence between $\boldsymbol{\mu}(\boldsymbol{\eta}_k)$ and $\boldsymbol{\mu}(\boldsymbol{\eta}_{\text{out}})$. It can be observed that $D$ will converge to 0 as $\alpha \rightarrow 0$. This indicates that the performance of our method can be guaranteed by a sufficiently discriminative feature space where the averaged Bergman divergence between ID-class means and OOD data mean is sufficiently large. This theory is empirically justified by our results in Section A.5 where CIDER are more beneficial to our method than SupCon with the former learning more powerful feature representations than the latter.

*Proof.* First, notice that $\hat{p}_{\boldsymbol{\theta}}(\mathbf{z}) \in [0, 1], \forall \mathbf{z} \in \mathcal{Z}$, Therefore, by Lemma 1, we have,

$$\mathbb{E}_{\mathbf{z} \in \mathcal{P}_{\text{in}}}[\hat{p}_{\boldsymbol{\theta}}(\mathbf{z})] - \mathbb{E}_{\mathbf{z} \in \mathcal{P}_{\text{out}}}[\hat{p}_{\boldsymbol{\theta}}(\mathbf{z})] \leq \delta(\mathcal{P}_{\text{in}}, \mathcal{P}_{\text{out}}) \tag{24}$$

Next, recall that $p_{\text{in}}(\mathbf{z})) = \frac{1}{K} \sum_{k=1}^{K} p_{\text{in}}(\mathbf{z}|k)$, let $\mathcal{P}_{\text{in}}^k$ denotes the probability distribution corresponding to $p_{\text{in}}(\cdot|k)$ and by triangle inequality and the definition of total variation we obtain

$$\delta(\mathcal{P}_{\text{in}}, \mathcal{P}_{\text{out}}) = \delta(\frac{1}{K} \sum_{k=1}^{K} \mathcal{P}_{\text{in}}^k, \mathcal{P}_{\text{out}}) = \sup_{A \in \mathcal{B}} \left| \frac{1}{K} \sum_{k=1}^{K} \mathcal{P}_{\text{in}}^k(A) - \mathcal{P}_{\text{out}}(A) \right| \tag{25}$$

$$\leq \frac{1}{K} \sum_{k=1}^{K} \sup_{A \in \mathcal{B}} \left| \mathcal{P}_{\text{in}}^k(A) - \mathcal{P}_{\text{out}}(A) \right| \tag{26}$$

$$= \frac{1}{K} \sum_{k=1}^{K} \delta(\mathcal{P}_{\text{in}}^k - \mathcal{P}_{\text{out}}) \tag{27}$$

Finally, by Lemma 2 and Lemma 3, we have:

$$\delta(\mathcal{P}_{\text{in}}^k - \mathcal{P}_{\text{out}}) \leq \sqrt{\frac{1}{2} KL(\mathcal{P}_1 \| \mathcal{P}_2)} = \sqrt{\frac{1}{2} d_\varphi(\boldsymbol{\mu}(\boldsymbol{\eta}_k), \boldsymbol{\mu}(\boldsymbol{\eta}_{\text{out}}))} \tag{28}$$

Putting all together, we obtain

$$\mathbb{E}_{\mathbf{z} \in \mathcal{P}_{\text{in}}}[\hat{p}_{\boldsymbol{\theta}}(\mathbf{z})] - \mathbb{E}_{\mathbf{z} \in \mathcal{P}_{\text{out}}}[\hat{p}_{\boldsymbol{\theta}}(\mathbf{z})] \leq \frac{1}{K} \sum_{k=1}^{K} \sqrt{\frac{1}{2} d_\varphi(\boldsymbol{\mu}(\boldsymbol{\eta}_k), \boldsymbol{\mu}(\boldsymbol{\eta}_{\text{out}}))} \tag{29}$$

and the proof is complete. $\qquad\square$

### A.11 CONTRIBUTION SUMMARY

The contributions of our method are summarised as follows:

- It is always non-trivial to generalize from a specific distribution/distance to a broader distribution/distance family since this will trigger an important question to the optimal design of the underlying distribution (♣). To answer this question, we explore the conjugate relationship as a guideline for the design. Compared with other hand-crafted choices, our proposed $l_p$ norm is general and well-defined, offering simplicity in determining its conjugate pair. By searching the optimal value of p for each dataset, we can flexibly model ID data in a data-driven manner instead of blindly adopting a narrow Gaussian distributional assumption in prior work, i.e., GEM (Morteza & Li, 2022) and Maha (Lee et al., 2018).

Table 9: Additional results of long-tailed OOD detection on Cifar-100, where we consider two baselines: (a) the ID training data is with class imbalance and (b) the ID training data is with class balance. ↑ indicates larger values are better and vice versa.

| Baseline | Maha | | GEM | | KNN | | Ours | |
|---|---|---|---|---|---|---|---|---|
| | FPR95↓ | AUROC↑ | FPR95↓ | AUROC↑ | FPR95↓ | AUROC↑ | FPR95↓ | AUROC↑ |
| (a) | 71.76 | 75.22 | 66.82 | 76.97 | 58.11 | 81.75 | 52.00 | 82.86 |
| (b) | 67.39 | 77.16 | 56.67 | 82.93 | 60.11 | 79.22 | 48.46 | 84.02 |

- Our proposed framework reveals the core components in density estimation for OOD detection, which was overlooked by most heuristic-based OOD papers. In this way, The framework not only inherits prior work including GEM (Morteza & Li, 2022) and Maha (Lee et al., 2018) but also motivates further work to explore more effective designing principles of density functions for OOD detection.

- We demonstrate the superior performance of our method on several OOD detection benchmarks (CIFAR10/100 and ImageNet-1K), different model architectures (DenseNet, ResNet, and MobileNet), and different pre-training protocols (standard classification, long-tailed classification and contrastive learning).

## A.12 LIST OF ASSUMPTIONS

The assumptions made in our method are given as follows:

1. The ID class prior is uniform, i.e., $\hat{p}_{\boldsymbol{\theta}}(k) = \frac{1}{K}$.

2. $g_{\varphi}(\cdot) = const$ and $\psi(\cdot) = \frac{1}{2}\|\|\cdot\|\|_p^2$

We note that 1) Assumption 1 is made in many post-hoc OOD detection methods either explicitly or implicitly (Jiang et al., 2023). Experiments in Section 4.4.2 show that our method still outperforms in long-tailed scenarios with Assumption 1, and 2) Assumption 2 helps to reduce the complexity of the exponential family distribution. While it is possible to parameterize the exponential family distribution in a more complicated manner, our proposed simple version suffices to perform well.

## A.13 A CLOSER LOOK AT EXPERIMENTS ON LONG-TAILED OOD DETECTION

As shown in Table 9, all methods that involve the use of ID training data suffer from a decrease in their averaged OOD detection performance when the ID training data is with class imbalance. Note that we keep using the network pre-trained on the long-tailed version of CIFAR-100 for fair comparison. Even so, our method consistently outperforms in both scenarios, which implies the robustness of our method. We suspect the reason is that the flexibility of the norm coefficient provides us with the chance to find a compromised distribution from the exponential family.

Table 10: Detailed results on six common OOD benchmark datasets: Textures, SVHN, Places365, LSUN-Crop, LSUN-Resize, and iSUN. For each ID dataset, we use the same DenseNet pretrained on CIFAR-100. ↑ indicates larger values are better and vice versa.

| Method | SVHN | | LSUN-C | | LSUN-R | | iSUN | | Texture | | Places365 | | Average | |
|---|---|---|---|---|---|---|---|---|---|---|---|---|---|---|
| | FPR95↓ | AUROC↑ | FPR95↓ | AUROC↑ | FPR95↓ | AUROC↑ | FPR95↓ | AUROC↑ | FPR95↓ | AUROC↑ | FPR95↓ | AUROC↑ | FPR95↓ | AUROC↑ |
| MSP | 81.70 | 75.40 | 60.49 | 85.60 | 85.24 | 69.18 | 85.99 | 70.17 | 84.79 | 71.48 | 82.55 | 74.31 | 80.13 | 74.36 |
| ODIN | 41.35 | 92.65 | 10.54 | 97.93 | 65.22 | 84.22 | 67.05 | 83.84 | 82.34 | 71.48 | 82.32 | 76.84 | 58.14 | 84.49 |
| Energy | 87.46 | 81.85 | 14.72 | 97.43 | 70.65 | 80.14 | 74.54 | 78.95 | 84.15 | 71.03 | 79.20 | 77.72 | 68.45 | 81.19 |
| React | 83.81 | 81.41 | 25.55 | 94.92 | 60.08 | 87.88 | 65.27 | 86.55 | 77.78 | 78.95 | 82.65 | 74.04 | 62.27 | 84.47 |
| DICE | 54.65 | 88.84 | 0.93 | 99.74 | 49.40 | 91.04 | 48.72 | 90.08 | 65.04 | 76.42 | 79.58 | 77.26 | 49.72 | 87.23 |
| ASH | 25.02 | 95.76 | 5.52 | 98.94 | 51.33 | 90.12 | 46.67 | 91.30 | 34.02 | 92.35 | 85.86 | 71.62 | 41.40 | 90.02 |
| Maha | 22.44 | 95.67 | 68.90 | 86.30 | 23.07 | 94.20 | 31.38 | 93.21 | 62.39 | 79.39 | 92.66 | 61.39 | 55.37 | 82.73 |
| SHE | 41.98 | 91.02 | 1.01 | 99.66 | 77.63 | 74.14 | 72.36 | 76.42 | 60.05 | 76.49 | 74.94 | 77.90 | 54.66 | 82.60 |
| KNN | 17.67 | 96.40 | 31.62 | 92.85 | 46.95 | 90.65 | 39.41 | 39.41 | 24.73 | 93.44 | 88.72 | 67.19 | 41.52 | 88.75 |
| Ours | 5.37 | 97.05 | 15.94 | 96.97 | 20.34 | 96.11 | 18.77 | 96.36 | 20.41 | 94.77 | 80.29 | 73.99 | 28.27 | 92.50 |
| Ours+ASH | 10.53 | 97.83 | 12.15 | 97.85 | 21.34 | 96.93 | 18.27 | 97.41 | 19.40 | 94.73 | 72.27 | 74.51 | 25.66 | 93.21 |

Table 11: Detailed results on six common OOD benchmark datasets: Textures, SVHN, Places365, LSUN-Crop, LSUN-Resize, and iSUN. For each ID dataset, we use the same DenseNet pretrained on CIFAR-10. ↑ indicates larger values are better and vice versa.

| Method | SVHN | | LSUN-C | | LSUN-R | | iSUN | | Texture | | Places365 | | Average | |
|---|---|---|---|---|---|---|---|---|---|---|---|---|---|---|
| | FPR95↓ | AUROC↑ | FPR95↓ | AUROC↑ | FPR95↓ | AUROC↑ | FPR95↓ | AUROC↑ | FPR95↓ | AUROC↑ | FPR95↓ | AUROC↑ | FPR95↓ | AUROC↑ |
| MSP | 47.24 | 93.48 | 33.57 | 95.54 | 42.10 | 94.51 | 42.31 | 94.52 | 64.15 | 88.15 | 63.02 | 88.57 | 48.73 | 92.46 |
| ODIN | 25.29 | 94.57 | 4.70 | 98.86 | 3.09 | 99.02 | 3.98 | 98.90 | 57.50 | 82.38 | 52.85 | 88.55 | 24.57 | 93.71 |
| Energy | 40.61 | 93.99 | 3.81 | 99.15 | 9.28 | 98.12 | 10.07 | 98.07 | 56.12 | 86.43 | 39.40 | 91.64 | 26.55 | 94.57 |
| DICE | 25.99 | 95.90 | 0.26 | 99.92 | 3.91 | 99.20 | 4.36 | 99.14 | 41.90 | 88.18 | 48.59 | 89.13 | 20.83 | 95.24 |
| React | 41.64 | 93.87 | 5.96 | 98.84 | 11.46 | 97.87 | 12.72 | 97.72 | 43.58 | 92.47 | 43.31 | 91.03 | 26.45 | 94.67 |
| ASH | 6.51 | 98.65 | 0.90 | 99.73 | 4.96 | 98.92 | 5.17 | 98.90 | 24.34 | 95.09 | 48.45 | 88.34 | 15.05 | 96.61 |
| Maha | 6.42 | 98.31 | 56.55 | 86.96 | 9.14 | 97.09 | 9.78 | 97.25 | 21.51 | 92.15 | 85.14 | 63.15 | 31.42 | 89.15 |
| SHE | 30.02 | 94.48 | 0.58 | 99.85 | 9.03 | 98.23 | 10.49 | 98.05 | 51.10 | 83.52 | 38.32 | 92.39 | 23.26 | 94.40 |
| KNN | 7.64 | 98.52 | 0.81 | 99.76 | 15.24 | 97.35 | 13.77 | 97.65 | 27.09 | 95.09 | 40.00 | 92.06 | 17.43 | 96.74 |
| Ours | 5.37 | 98.95 | 7.13 | 98.63 | 5.18 | 98.92 | 6.42 | 98.65 | 17.06 | 96.72 | 42.37 | 91.82 | 13.92 | 97.15 |
| Ours+ASH | 6.61 | 98.70 | 1.87 | 99.56 | 1.65 | 99.58 | 2.77 | 99.39 | 17.22 | 96.83 | 47.72 | 91.83 | 12.14 | 97.65 |

Table 12: Detailed results on six common OOD benchmark datasets: Textures, SVHN, Places365, LSUN-Crop, LSUN-Resize, and iSUN. For each ID dataset, we use the same DenseNet pretrained on CIFAR-100. ↑ indicates larger values are better and vice versa.

| Method | SVHN | | LSUN-C | | LSUN-R | | iSUN | | Texture | | Places365 | | Average | |
|---|---|---|---|---|---|---|---|---|---|---|---|---|---|---|
| | FPR95↓ | AUROC↑ | FPR95↓ | AUROC↑ | FPR95↓ | AUROC↑ | FPR95↓ | AUROC↑ | FPR95↓ | AUROC↑ | FPR95↓ | AUROC↑ | FPR95↓ | AUROC↑ |
| MSP | 63.14 | 58.98 | 91.36 | 58.71 | 75.09 | 81.99 | 78.36 | 79.93 | 87.50 | 73.10 | 87.01 | 69.27 | 79.29 | 79.29 |
| ODIN | 88.46 | 77.68 | 77.68 | 90.50 | 72.78 | 84.61 | 76.01 | 83.04 | 87.61 | 74.81 | 87.30 | 69.78 | 77.47 | 79.90 |
| Energy | 40.61 | 93.99 | 3.81 | 99.15 | 9.28 | 98.12 | 10.07 | 98.07 | 56.12 | 86.43 | 39.40 | 91.64 | 26.55 | 94.57 |
| DICE | 86.75 | 78.47 | 18.07 | 95.95 | 79.00 | 81.75 | 81.76 | 81.16 | 81.13 | 75.84 | 91.65 | 65.97 | 73.06 | 79.86 |
| React | 91.73 | 78.45 | 40.08 | 90.98 | 67.93 | 84.65 | 66.36 | 85.30 | 68.60 | 81.60 | 84.99 | 70.96 | 69.95 | 81.99 |
| ASH | 59.22 | 88.20 | 26.21 | 95.37 | 77.01 | 77.03 | 79.35 | 76.27 | 61.13 | 84.47 | 84.47 | 66.83 | 65.29 | 81.36 |
| Maha | 57.16 | 58.98 | 91.36 | 58.71 | 74.26 | 65.38 | 71.69 | 65.45 | 53.40 | 69.04 | 82.65 | 61.31 | 71.75 | 63.14 |
| KNN | 46.63 | 89.84 | 60.87 | 84.50 | 65.13 | 87.76 | 65.49 | 86.56 | 54.15 | 85.65 | 81.04 | 70.98 | 62.22 | 84.22 |
| Ours | 31.35 | 93.26 | 40.53 | 91.10 | 68.25 | 87.54 | 67.51 | 86.97 | 42.41 | 88.94 | 76.77 | 75.77 | 54.47 | 87.26 |

