# OpenReview forum: "ConjNorm: Tractable Density Estimation for Out-of-Distribution Detection"
_ICLR.cc/2024/Conference — ICLR 2024 poster_

### Official Review · Reviewer_MqR8 · 2023-10-30

**Soundness:** 3 good
**Presentation:** 3 good
**Contribution:** 2 fair
**Rating:** 5
**Confidence:** 4

**Summary:**

This paper proposes a new scoring method for post-hoc out-of-distribution (OOD) detection, by considering the OOD detection problem as a density estimation over the exponential family. Using the connections between the exponential family of distributions and the Bregman divergence, the original density estimation problem over the exponential family is converted into the problem of finding the Bregman divergence. Then, to reduce the search space for selecting Bregman divergence, the authors propose a pair of conjugate functions and reframe the original problem into the problem of finding the optimal norm coefficient $p$ against the given dataset. The partition function is estimated using the Mont Carlo-based importance sampling technique. Experimental results demonstrate the efficacy of the proposed score, with varying $p$ depending on the dataset (p=2.2 for CIFAR-10, 2.5 for CIFAR-100, 1.5 and 1.8 for ImageNet-1k on ResNet50 and MobileNetv2).

**Strengths:**

- Proposed a unified scoring method for post-hoc OOD detection using a general exponential family.
- Converted the original search problem over the expansive function space of Bregman divergence into a simple problem of selecting optimal norm coefficient.
- Demonstrated the effectiveness of the proposed score on CIFAR-10/100, ImageNet-1k, and scenarios including hard OOD detection and long-tailed OOD detection.

**Weaknesses:**

- The main search problem of optimal coefficient $p$ for OOD scoring is remained as a hyperparameter search, which may constrain the practicality of the proposed score. Furthermore, the OOD detection performance (FPR95) is quite sensitive to the value of $p$ (Figure 4), which implies that the performance of the proposed score highly depends on the hyperparameter $p$.

**Questions:**

- Can the authors provide any reasonable method to choose $p$ given the training dataset and the corresponding features given NN architecture without the hyperparameter search?
- The partition function is estimated using the importance sampling-based approximation. For long-tailed OOD detection when the ID training data exhibits an imbalanced class distribution, I guess the accuracy of the importance sampling-based estimation may decrease given a limited number of tail-class predictions. Given that, can the authors elaborate why their method still outperforms in the long-tailed scenarios?

---

> ### Author Response · Authors · 2023-11-17
> **Response to Reviewer MqR8**
>
> We thank reviewer MqR8 for your valuable comments. We updated the submission accordingly. Please kindly find the detailed responses below.
>
> > Q1: The main search problem of optimal coefficient for OOD scoring is a hyperparameter search, which may constrain the practicality of the proposed score, and How to choose $p$.
>
> **A1**: Similar to DICE [d], we adopt a validation set of Tiny ImageNet as the auxiliary OOD data for tuning $p$. we empirically define the search space of $p$ as (1,3]. We also observe that SOTA post-hoc OOD detection methods [a,b,c,d,e,f] also come with (one or more) hyper-parameters, where their searched values vary across datasets as well.
>
> We would like to argue that the tuning $p$ is reasonable and necessary for density-based OOD detection since the (latent feature) distribution of different datasets could not be necessarily the same as each other. By simply adjusting the value of the norm coefficient, we can succeed in finding the (relatively) most suitable distribution from the exponential family for each ID dataset in a computationally efficient manner, which is exactly one of the strengths of our method.
>
> [a] Extremely simple activation shaping for out-of-distribution detection. ICLR 2023
>
> [b] LINe: Out-of-Distribution Detection by Leveraging Important Neurons. CVPR 2023
>
> [c] Out-of-distribution detection with deep nearest neighbours. ICML 2022.
>
> [d] Dice: Leveraging sparsification for out-of-distribution detection. ECCV 2022
>
> [f] Out-of-Distribution Detection via Conditional Kernel Independence Model. NeurIPS 2022
>
> > Q2: For long-tailed OOD detection when the ID training data exhibits an imbalanced class distribution, I guess the accuracy of the importance sampling-based estimation may decrease given a limited number of tail-class predictions. Elaborate why their method still outperforms in the long-tailed scenarios.
>
> **A2**:  We agree with your insightful intuition. The mentioned class imbalance is widely recognized as a challenging setting. In response to your valuable comments, we have conducted the following experiments.
>
> As shown in the Table below, all methods that involve the use of ID training data suffer from a decrease in their averaged OOD detection performance when the ID training data is with class imbalance. Note that we keep using the network pre-trained on the long-tailed version of CIFAR-100 for fair comparison. Even so, our method consistently outperforms in both scenarios, which implies the robustness of our method. We suspect the reason is that the flexibility of the norm coefficient provides us with the chance to find a compromised distribution from the exponential family.
>
> |                |      Maha    |              |      KNN     |              |      GEM     |              |       Ours      |              |
> |:--------------:|:------------:|:------------:|:------------:|:------------:|:------------:|:------------:|:---------------:|:------------:|
> |                |     FPR95    |     AUROC    |     FPR95    |     AUROC    |     FPR95    |     AUROC    |       FPR95     |     AUROC    |
> |       class imbalance      |     71.76    |     75.22    |     58.11    |     81.75    |     66.82    |     76.97    |      52.00     |     82.86    |
> |       class balance     |     67.39    |     77.16    |     56.67    |     82.93    |     60.11    |     79.22    |      48.46     |     84.02    |
>
> Thanks again for your valuable comments. We have revised the submission accordingly by adding the discussion above. Please refer to Section A.12 in the revision for details.

---

> > ### Author Response · Authors · 2023-11-21
> > **Welcome for More Discussions**
> >
> > Dear Reviewer MqR8,
> >
> > Thanks for your valuable time in reviewing and insightful comments. Following your comments, we have tried our best to provide responses and revise our paper. Here is a **summary of our response** for your convenience:
> >
> > - **Hyperparameter Searching**: According to your insightful comment, we have provided a detailed illustration on how we choose $p$.
> > - **More Analysis on Long-tailed OOD Detection**: We also provided the more experiment results to analysis influence of class-imbalanced ID training data when they are used in post-hoc OOD. In particular, all methods that involve the use of ID training data suffer from a decrease in their averaged OOD detection performance when the ID training data is with class imbalance while our method consistently outperforms in both scenarios.
> >
> > We humbly hope our repsonse has addressed your concerns. If you have any additional concerns or comments that we may have missed in our responses, we would be most grateful for any further feedback from you to help us further enhance our work.
> >
> > Best regards
> >
> > The authors

---

> > > ### Author Response · Authors · 2023-11-22
> > > **Window for Discussion and Revision is Closing**
> > >
> > > Dear Reviewer MqR8,
> > >
> > > Thanks a lot for your time in reviewing and insightful comments, according to which we have carefully revised the paper to answer the questions. We sincerely understand you’re busy. But since the discussion due is approaching, would you mind checking the response and revision to confirm where you have any further questions?
> > >
> > > We are looking forward to your reply and happy to answer your further questions.
> > >
> > > Best regards
> > >
> > > Authors

---

> > > > ### Comment · Reviewer_MqR8 · 2023-11-22
> > > >
> > > > I appreciate the authors for providing the detailed response. After reading the authors' response, I am still a bit concerned about the practicality of the proposed score in the process of choosing the optimal coefficient $p$ for OOD scoring, The hyperparameter $p$ is a core part of the proposed OOD scoring, and the OOD detection performance (FPR95) highly depends on the value of $p$. The authors commented that they adopt a validation set of Tiny ImageNet as the auxiliary OOD data for tuning $p$, but this 1) requires additional auxiliary OOD data and 2) makes the chosen $p$ to depend not only on the in-distribution data but also on the auxiliary OOD data, which is not desirable in general. I think it will be great if the authors can provide any alternative approach in choosing $p$ without using the auxiliary OOD data.

---

> ### Author Response · Authors · 2023-11-22
> **Thanks for your swift reply and insightful comments**
>
> We're grateful for your quick feedback during this busy period. We would like to emphasize that
> 1) it is important and necessary to tune $p$ indeed since it is reasonable for the ID distribution $p(x)$ to vary from dataset to dataset. We kindly note that previous methods impose a fixed distributional assumption for all datasets (Gaussian in GEM and Maha; Gibbs-Boltzmann in Energy). However, different datasets should posse a different $p(x)$, based on the optimal results achieved: p=2.2 for CIFAR-10, 2.5 for CIFAR-100, 1.5 and 1.8 for ImageNet-1k on ResNet50 and MobileNetv2.
>
> 2) while the OOD detection performance depends on the value of $p$, it can be seen from Figures 4.c and 4.c that the performance of our method remains comparable to SOTA in CIFAR-10 when $p \in [1.6,2.4]$ and consistently outperforms SOTA in CIFAR-100 when $p \in [1.8,3.0]$.
>
> We agree with the reviewer that currently searching for the optimal p is suboptimal as the auxiliary data is required for cross-validation. It is definitely more practical if an OOD detector can adjust its hyper-parameter(s) with only access to its ID data though this is still an open problem in the OOD detection domain.
>
> Inspired by your insightful comments, we consider one of our future works to synthesize OOD samples with powerful generative models such as stable-diffusion models as alternative sources for auxiliary OOD validation data. The corresponding exploration will be conducted very soon.

---

### Official Review · Reviewer_LYsh · 2023-11-01

**Soundness:** 3 good
**Presentation:** 3 good
**Contribution:** 2 fair
**Rating:** 6
**Confidence:** 3

**Summary:**

The paper studies the task of out-of-distribution (OOD) data detection for supervised learning tasks. The proposed method is essentially along the lines of density level set thresholding, and the density estimates leverage a particular type of exponential family (uniform) mixtures originating from a Bregman-divergence-related framework. After choosing specific parameters for computational tractability and performance, the authors compared their method with multiple baselines and prior works to demonstrate its effectiveness.

**Strengths:**

The paper reads well and introduces background and prior works properly. In addition, the authors leverage concrete examples and visualizations, such as plots and tables, to help the reader get the most critical points. All these benefit the readability and clarity of the submission.

Regarding novelty and originality, the proposed method differs from existing approaches algorithmically and originates from a more general framework. It is good that the authors spend efforts building theoretical justifications and showing underlying motivations.

The experiment results (seem to) suggest that the new method has advantages over the prior works and often outperforms its predecessors, at least within the experimental setting of the authors. Sometimes, there are seven to ten competitors and two to five benchmark datasets, such as ImageNet and CIFAR-10. From my perspective, this is the most substantial contribution added by the paper, showing the method's practicality.

**Weaknesses:**

One area for improvement is that while the method comes from a theoretical framework, I need to find explicit theoretical guarantees and technical claims to justify its effectiveness. So, the lack of theoretical justification is a weakness worth addressing, potentially deriving some for simple cases like the Gaussian one or explaining why the algorithm tends to perform well for small p values.

The second weakness is that the prior work (Morteza & Li, 2022) already proposed a method based on Gaussian assumptions and Mahalanobis distance. The extension in this paper, at least logically, is relatively straightforward, i.e., from Gaussian to Exponential Family and from Mahalaobis distance to Bregman-divergence (an extension). Maybe it's worth adding a section summarizing the paper's technical novelty.

Another weakness is that while the framework's formulation seems general, multiple assumptions come along the way. For example, the authors assumed a uniform prior and set $\psi$ to the $l_p$ norm. Of course, these might be necessary for a computationally tractable approach and could be acceptable in their current form.

**Questions:**

Following the comments on the weaknesses, it would be helpful if the authors could
- Provide theoretical justification (guarantees) for the proposed method.
- Explain and summarize the technical novelty in addition to prior works.
- List out all the assumptions made leading to the final method.

In addition, it would be good if the authors could make the notations more distinguishable, e.g., addressing the overuse of $\hat{p}_\theta(\star)$.

---

> ### Author Response · Authors · 2023-11-17
> **Response to Reviewer LYsh (Part 1)**
>
> We thank Reviewer LYsh for your thorough comments. As to the weaknesses and minor issues you pointed out, we took them very seriously, and have updated parts of the paper to improve it. Our response is as follows.
>
> > Q1.1: theoretical justification
>
> **A1.1**:
>
> **Setup.** Let $\mathcal{P}\_{\text{in}}$ and $\mathcal{P}\_{\text{out}}$ are the underlying distributions for ID and OOD data. We use $p\_{\text{in}} (\mathbf{z})$ and $p\_{\text{out}} (\mathbf{z})$ to denote the probability density function where the input $\mathbf{z}$ is sampled from the feature embeddings space $\mathcal{Z}$. We model ID data as a mixture of ID-class conditioned exponential family distributions, i.e.,
> \begin{equation}
>     p\_{\text{in}}\left(\mathbf{z}\right) = \frac{1}{K} \sum\_{k=1}^{K} p\_{\text{in}}\left(\mathbf{z} | k\right) = \frac{1}{K} \sum\_{k=1}^{K} \frac{\exp(-d\_\varphi(\mathbf{z},\boldsymbol{\mu}(\boldsymbol{\eta}\_k)))}{\int \exp(-d\_\varphi(\mathbf{z}',\boldsymbol{\mu}(\boldsymbol{\eta}\_k))){\rm d}\mathbf{z}'}
> \end{equation}
> Inspired by open-set recognition [a], we treat OOD data as a whole and model it as a single exponential family distribution parameterized by $\boldsymbol{\eta}\_{\text{out}}$, i.e.,
> \begin{equation}
>     p\_{\text{out}}\left(\mathbf{z}\right)=\frac{\exp(-d\_\varphi(\mathbf{z},\boldsymbol{\mu}(\boldsymbol{\eta}\_\text{out})))}{\int \exp(-d\_\varphi(\mathbf{z}',\boldsymbol{\mu}(\boldsymbol{\eta}\_\text{out}))){\rm d}\mathbf{z}'}.
> \end{equation}
> We consider the following measure to how well our method distinguishes ID data from OOD data:
> \begin{equation}
>     D:=\mathbb{E}\_{\mathbf{z} \in \mathcal{P}\_{\text{in}}} [\hat{p}\_{\boldsymbol{\theta}}\left(\mathbf{z}\right)] - \mathbb{E}\_{\mathbf{z} \in \mathcal{P}\_{\text{out}}} [\hat{p}\_{\boldsymbol{\theta}}\left(\mathbf{z}\right)]
> \end{equation}
>
> **Main Result.** We have the following bound:
> \begin{equation}
>     \mathbb{E}\_{\mathbf{z} \in \mathcal{P}\_{\text{in}}} [\hat{p}\_{\boldsymbol{\theta}}\left(\mathbf{z}\right)] - \mathbb{E}\_{\mathbf{z} \in \mathcal{P}\_{\text{out}}} [\hat{p}\_{\boldsymbol{\theta}}\left(\mathbf{z}\right)] \le \alpha
> \end{equation}
> where $\alpha: = \frac{1}{K} \sum_{k=1}^{K} \sqrt{\frac{1}{2}d\_\varphi(\boldsymbol{\mu}(\boldsymbol{\eta}_k),\boldsymbol{\mu}(\boldsymbol{\eta}\_{\text{out}}))}$. Please refer Section A.10 in the revised manuscript for detailed proof.
>
> **Main Result** bounds the measure $D$ in terms of Bregman divergence between $\boldsymbol{\mu}(\boldsymbol{\eta}\_k)$ and $\boldsymbol{\mu}(\boldsymbol{\eta}\_{\text{out}})$. It can be observed that $D$ will converge to 0 as $\alpha \rightarrow 0$. This indicates that the performance of our method can be guaranteed by a sufficiently discriminative feature space where the averaged Bergman divergence between ID-class means and OOD data mean is sufficiently large. Note that this theory is empirically justified by our results in Section A.5 where CIDER are more beneficial to our method than SupCon with the former learning more powerful feature representations than the latter.
>
> In the future, we will delve deeper into the theoretical understanding of our method as our future work.
>
> [a]  Adversarial reciprocal points learning for open set recognition. TPAMI, 2021.
>
> > Q1.2: Explain why the algorithm tends to perform well for small p values
>
> **A1.2**: Since the searching process of the coefficient $p$ is data-driven, the optimal value of $p$ should vary from dataset to dataset. Therefore, while extensive experiments show that small P values tend to be beneficial to OOD detection on CIFAR and ImageNet, this observation does not necessarily hold for all datasets..

---

> ### Author Response · Authors · 2023-11-17
> **Response to Reviewer LYsh (Part 2)**
>
> > Q2: Maybe it's worth adding a section summarizing the paper's technical novelty.
>
> **A2**: We thank you for your advice. The contributions of our method are summarised as follows:
>
> - It is always non-trivial to generalize from a specific distribution/distance to a broader distribution/distance family since this will trigger an important question to the optimal design of the underlying distribution ($\clubsuit$).To answer this question, we explore the conjugate relationship as a guideline for the design. Compared with other hand-crafted choices, our proposed $l_p$ norm is general and well-defined, offering simplicity in determining its conjugate pair. By searching the optimal value of p for each dataset, we can flexibly model ID data in a data-driven manner instead of blindly adopting a narrow Gaussian distributional assumption in prior work, i.e., GEM and Maha.
>
> - Our proposed framework reveals the core components in density estimation for OOD detection, which was overlooked by most heuristic-based OOD papers. In this way, The framework not only inherits prior work including GEM and Maha but also motivates further work to explore more effective designing principles of density functions for OOD detection.
>
> - We demonstrate the superior performance of our method on several OOD detection benchmarks (CIFAR10/100 and ImageNet-1K), different model architectures (DenseNet, ResNet, and MobileNet), and different pre-training protocols (standard classification, long-tailed classification and Contrastive learning).
>
> We included the summary above in Section A.11 of the revision.
>
> > Q3: List out all the assumptions made
>
> **A3**: Thank you for your advice. The assumptions made in our method are given as follows:
>
> - **Assumption 1.** The ID class prior is uniform, i.e., $\hat{p}_{\boldsymbol{\theta}}\left(k\right)=\frac{1}{K}$.
>
> **Assumption 1** helps our method work without true knowledge of the ID class prior distribution. We note that the assumption is also made in many post-hoc OOD detection methods either explicitly or implicitly [b]. Experiments in Section 4.4.2 show that our method still outperforms in long-tailed scenarios.
>
> - **Assumption 2.**  $g_\varphi(\cdot)=const$ and $\psi(\cdot) = \frac{1}{2}\|\|\cdot\|\|_{p}^{2}$
>
> **Assumption 2** helps to reduce the complexity of the exponential family distribution. While it is possible to parameterize the exponential family distribution in a more complicated manner, our proposed simple version suffices to perform well.
>
> We listed the assumptions above in Section A.12 of the revision.
>
> [b] Detecting Out-of-distribution Data through In-distribution Class Prior. ICML 2023
>
> > Q4: it would be good if the authors could make the notations more distinguishable.
>
> **A4**. Thanks for your advice. We have improved our used notations in the revised version.

---

> > ### Author Response · Authors · 2023-11-21
> > **Welcome for More Discussions**
> >
> > Dear Reviewer LYsh,
> >
> > Thanks for your valuable time in reviewing and insightful comments. Following your comments, we have tried our best to provide responses and revise our paper. Here is a **summary of our response** for your convenience:
> >
> > - **Theoretical Guarantee**: Following your insightful comment, We add a section providing a theoretical guarantee for our method.
> > - **Contribution Summary**: According to your valuable advice, we add a section summarizing the paper's technical novelty in our revised manuscript.
> > - **Assumption List**: According to your constructive advice, we add a section summarizing the assumptions made in our method.
> >
> > We humbly hope our response has addressed your concerns. If you have any additional concerns or comments that we may have missed in our responses, we would be most grateful for any further feedback from you to help us further enhance our work.
> >
> > Best regards
> >
> > The authors

---

> > > ### Author Response · Authors · 2023-11-22
> > > **Window for Discussion and Revision is Closing**
> > >
> > > Dear Reviewer LYsh,
> > >
> > > Thanks a lot for your time in reviewing and insightful comments, according to which we have carefully revised the paper to answer the questions. We sincerely understand you’re busy. But since the discussion due is approaching, would you mind checking the response and revision to confirm where you have any further questions?
> > >
> > > We are looking forward to your reply and happy to answer your further questions.
> > >
> > > Best regards
> > >
> > > Authors

---

### Official Review · Reviewer_WvTp · 2023-11-08

**Soundness:** 3 good
**Presentation:** 3 good
**Contribution:** 3 good
**Rating:** 8
**Confidence:** 4

**Summary:**

The authors present a data density estimation method targeted towards out of distribution detection. The authors parameterize Bregman Divergences, which in turn are shown to parameterize Exponential Families up to an approximable normalization constant.

The authors conduct extensive evaluations on numerous OOD tasks and include ablations, showing improvements over benchmarks on many tasks.

In general the mathematical presentation is careful, results are contextualized, and the reader learns both about OOD in general and about the specific presented method, picking up some tricks on parameterizing densities along the way.

**Strengths:**

The authors conduct extensive evaluations on numerous OOD tasks and include ablations, showing improvements over benchmarks on many tasks.

In general the mathematical presentation is careful, results are contextualized, and the reader learns both about OOD in general and about the specific presented method, picking up some tricks on parameterizing densities along the way.

**Weaknesses:**

Nothing particularly bad and things seem correct and well reported and well explored.

Mostly, the weakness would just be the lack of discussion of Deep Generative Models. The paper seems to present density estimation as the main challenge of OOD detection. For some of the considered benchmarks such as GEM, the main criticism is the specific distribution assumptions, hence the exploration in this work across the Exponential Family.

On the other hand, Deep generative models (DGMs) are a flexible approach for modeling data distributions without making distributional assumptions. Not all DGMs give the user a computable density, but there are some that do such as Normalizing Flows and more recently methods like Flow Matching, Stochastic Interpolants, and others to name a few. More generally some models give you un-normalized log densities, which also seem to be okay for this work considering that this work is willing to estimate certain normalization constants.

It's totally okay to explore a non-DGM-based method in this work, but I think it would strongly benefit from some contextualization and an attempt to answer this question in at least one way:

For high dimensional data such as images, why should someone not pick generic deep generative models that admit densities (or un-normalized densities, or maybe log density lower bounds) and why instead should someone stick with search within the exponential family (for which you give good methods, algorithms, etc, and for which you get good results)?

There is some older work on role of DGMs in OOD detection (such as https://arxiv.org/abs/2107.06908) but I think a LOT of progress has been made in image DGMs since then (like DDPM https://arxiv.org/abs/2006.11239, Interpolants, https://arxiv.org/abs/2209.15571, Diffusions in latent space, https://arxiv.org/abs/2212.09748, etc)

**Questions:**

1)

Could you please clarify this phrase? I re-read it a few times and just did not understand its meaning

"Without loss of generality, we employ latent features z extracted from deep models as a surrogate for the original high-dimensional raw data x. This is because z is deterministic within the post-hoc framework."

2)

Please answer my main question in Weaknesses, on why no discuss of the role in DGMs for flexible density estimation in OOD detection.

3) small comment:
please tell the reader more about why learning the natural parameter of an Exp. Family intractable.

---

> ### Author Response · Authors · 2023-11-17
> **Response to Reviewer WvTp**
>
> We thank Reviewer WvTp for your constructive comments. To address some of your concerns, we have updated our paper. Please see below for our point-to-point rebuttal.
>
> > Q1: Clarify “Without loss of generality, we employ latent features z extracted from deep models as a surrogate for the original high-dimensional raw data $\mathbf{x}$. This is because $\mathbf{z}$ is deterministic within the post-hoc framework”
>
> **A.1**: Thanks for pointing out the potentially confusing description.
> - Since this paper mainly focuses on the feature space for determining OOD data, we use different notations to explicitly discriminate raw input data $\mathbf{x}$ from the latent feature $\mathbf{z}$ to make things clearer. Note that some prior papers sometimes use $\mathbf{z}$ and $\mathbf{x}$ interchangeably.
> - To clarify, the term "deterministic" here means that, for any given input image x, we can always have a deterministic representation z since the pre-trained encoder is fixed in the setting of post-hoc OOD detection. This motivates us to choose the lower-dimensional feature space as a suitable surrogate of the raw data space X for more computationally efficient density estimation.
>
> > Q2: Discuss the role of Deep generative models (DGMs) for flexible density estimation in OOD detection
>
> **A.2**: Thanks for your kind suggestion.
> - We agree that using DGMs for density estimation is, of course, a valid and intuitive option. However, aligning with our response in **A.1**, this practice requires training DGMs from scratch to reconstruct high-dimensional $\mathbf{x}$, therefore bringing more computational overheads.
>
> - Please kindly note that our paper focuses on the task of Post-hoc OOD detection where only pre-trained models at hand are expected to be used to detect OOD data from streaming data at the interference stage.
>
> - We also explore the possibility of integrating pre-trained Diffusion models [a,b] into zero-shot class-conditioned density estimation based on Eq.(1) in [c]. Unfortunately, the computation is intractable due to the integral. Although authors in [c] use a simplified ELBO for approximation, there is no theoretical guarantee that the ELBO can align with the data density not to mention the computational-inefficient inference of diffusion models. We will leave this challenge as our future work.
>
> [a] Scalable Diffusion Models with Transformers. ICCV 2023.
>
> [b] High-Resolution Image Synthesis with Latent Diffusion Models. CVPR 2022
>
> [c] Your Diffusion Model is Secretly a Zero-Shot Classifier. ICCV 2023.
>
> Thanks again for the valuable suggestion. Accordingly, we have added the discussion on DGMs for density estimation in Section A.8 of the revised version.
>
> > Q3: Tell more about why learning the natural parameter of an Exp. Family is intractable
>
> **A.3**: Thank you for your advice. given the fact that $\int \hat{p}_{\boldsymbol{\theta}}\left(\mathbf{z}|k \right) {\rm d}\mathbf{z} =1$, we then have:
>
> $$ \int \exp \left\lbrace \mathbf{z}^\top\boldsymbol{\eta}\_k-\psi(\boldsymbol{\eta}_k)-g\_{\psi}(\mathbf{z})\right\rbrace{\rm d}\mathbf{z}=1 $$
>
> This means that, for any known $\psi(\cdot)$ and $g\_{\psi}(\cdot)$, one can learn the natural parameter $\boldsymbol{\eta}_k$ by solving the following equation:
>
> $$\exp \psi(\boldsymbol{\eta}_k)=\int \exp \left\lbrace \mathbf{z}^\top\boldsymbol{\eta}\_k-g\_{\psi}(\mathbf{z})\right\rbrace{\rm d}\mathbf{z}$$
>
> Since the right side of the equation includes the integral over latent feature space that is high-dimensional, learning the natural parameter of an Exp. Family is said to be intractable.
>
> We add the elaboration above in Section A.9 of the revised version.

---

> > ### Author Response · Authors · 2023-11-21
> > **Welcome for More Discussions**
> >
> > Dear Reviewer WvTp,
> >
> > Thanks for your valuable time in reviewing and insightful comments. Following your comments, we have tried our best to provide responses and revise our paper. Here is a **summary of our response** for your convenience:
> >
> > - **Clarity**: We have provided an explanation about the potentially confusing phrase.
> > - **Related Work**: Following your constructive comments, we have discussed related works on deep generative models for density estiomation in our revised version, where we not only include how conventional technique, such as autoregressive and normalizing flows, used for density esitmation but also discuss the possibility and challenge of  applying pre-trained diffusion models for zero-shot density estiomation.
> > - **Illustration**: As per your valueab comments, we have provided a detailed discussion on why learning the natural parameter of an Exp. Family is intractable. In short, the intracatbility arises from the integral over latent feature space that is high-dimensional during the computation.
> >
> > We humbly hope our repsonse has addressed your concerns. If you have any additional concerns or comments that we may have missed in our responses, we would be most grateful for any further feedback from you to help us further enhance our work.
> >
> > Best regards
> >
> > The authors

---

> ### Author Response · Authors · 2023-11-22
> **Window for Discussion and Revision is Closing**
>
> Dear Reviewer WvTp,
>
> Thanks a lot for your time in reviewing and insightful comments, according to which we have carefully revised the paper to answer the questions. We sincerely understand you’re busy. But since the discussion due is approaching, would you mind checking the response and revision to confirm where you have any further questions?
>
> We are looking forward to your reply and happy to answer your further questions.
>
> Best regards
>
> Authors

---

### Official Review · Reviewer_EBSh · 2023-11-09

**Soundness:** 2 fair
**Presentation:** 3 good
**Contribution:** 2 fair
**Rating:** 6
**Confidence:** 2

**Summary:**

This paper studies the problem of density estimation for density-based out-of-distribution detection. The authors firstly point out that existing logit, distance, and density based OOD methods can be either inconsistent or too restrictive. Then the authors utilize the property of exponential family and its relation to Bregman divergence to induce a modeling of the density function using conjugate norms. They further combine different method for the estimation of partition functions and finally experimentally validate their methods.

**Strengths:**

1. The assumption of exponential family is mild and the authors utilize the property of conjugate functions to derive a concise formulation of density functions.
2. The estimation of partition functions can be combined with different methods, which indicates the flexibility of the proposed method.
3. The experimental results on widely-used benchmark datasets validate the usefulness of the proposed method.

**Weaknesses:**

The theoretical results of this paper are based on the exponential family of distribution. I think this is an explicit assumption on the prior distribution, which contradicts your answer to question ♠. An analysis on the potential extension of your results can be helpful.

**Questions:**

Please see the weaknesses.

---

> ### Author Response · Authors · 2023-11-17
> **Response to Reviewer EBSh**
>
> We appreciate the insightful comments provided by Reviewer EBSh. Please see our responses to your concerns below.
>
> > Q1: I think this is an explicit assumption on the prior distribution, which contradicts $\spadesuit$ ( how can we obtain a tractable estimate for $\Phi (k)$ without presuming any particular prior distribution of $\hat{p}_{\boldsymbol{\theta}}\left(\mathbf{z}|k \right)$ )
>
> **A.1**: Thanks for your insightful comments. We would like to note that: 1) the mild assumption of the exponential family is introduced to guide the design of density function ($\clubsuit$), where we search against the given dataset for the best choice of $l_p$ to determine the optimal density function. 2) $\spadesuit$ targets at the computation of the normalizing constant $\Phi(k)$ in Eq.(11) that involves the integral over high-dimensional feature space. Different from prior work that specifies $\hat{p}_{\boldsymbol{\theta}}\left(\mathbf{z}|k \right)$ as a pre-defined distribution, e.g., Guassian, to simplify $\Phi(k)$ as a known value, we alternatively design an importance sampling-based estimator of $\Phi (k)$ as the solution to $\spadesuit$ without loss of generalization. Note that our proposed estimator does not rely on any prior knowledge of data distribution and therefore can be ideally applied to any forms of density functions.

---

> > ### Author Response · Authors · 2023-11-21
> > **Welcome for More Discussions**
> >
> > Dear Reviewer EBSh,
> >
> > Thanks for your valuable time in reviewing and insightful comments. Following your comments, we have tried our best to provide responses and revise our paper. Here is a **summary of our response** for your convenience:
> >
> > - **Explanation of $\spadesuit$**: According to your insightful question, we have provided a detailed discussion on the reason why an explicit assumption on the prior distribution does not contradict $\spadesuit$
> >
> > We humbly hope our repsonse has addressed your concerns. If you have any additional concerns or comments that we may have missed in our responses, we would be most grateful for any further feedback from you to help us further enhance our work.
> >
> > Best regards
> >
> > The authors

---

> > > ### Author Response · Authors · 2023-11-22
> > > **Window for Discussion and Revision is Closing**
> > >
> > > Dear Reviewer EBSh,
> > >
> > > Thanks a lot for your time in reviewing and insightful comments, according to which we have carefully revised the paper to answer the questions. We sincerely understand you’re busy. But since the discussion due is approaching, would you mind checking the response and revision to confirm where you have any further questions?
> > >
> > > We are looking forward to your reply and happy to answer your further questions.
> > >
> > > Best regards
> > >
> > > Authors

---

### Author Response · Authors · 2023-11-17
**General Response**

Dear Area Chairs and Reviewers,

We would like to thank the reviewers again for their constructive and insightful comments, which help us a lot in improving the submission. We have uploaded the revised version and responded to all the reviewers in detail. We believe that the quality of the paper is improved and the contributions are solid. In particular, we would like to highlight some key materials we added:

1. Discussion on deep generative models as our related work (Section A.8)
2. Elaboration on why learning the natural parameter of an Exp. Family is intractable (Section A.9)
3. Theoretical justification (Section A.10)
4. Contribution summary (Section A.11)
5. List of assumptions made in our method (Section A.12)
6. The searching strategy of the norm coefficient $p$ (Section A.3)
7. Discussion on experiment results of long-tailed OOD detection (Section A.13)

We understand that reviewers are busy during the response period, we would greatly appreciate it if the reviewers could kindly advise if our responses solve their concerns. If there are any other suggestions/questions, we will try our best to provide satisfactory answers. We are looking forward to any further discussion with the reviewers. Thank you for your time.

Best regards,

The authors

---

### Meta-Review · Area_Chair_zzPD · 2023-12-02

**Metareview:**

Reviewers generally agreed that the paper makes novel and interesting theoretical contributions to OOD detection, by presenting an exponential family based approach to estimating densities in a tractable manner. The paper is generally written clearly and provides a nice framework for reasoning about the problem.

One reviewer raised a concern about the paper's sensitivity to the parameter $p$ (controlling the norm), which appears to require some OOD validation data. The response argued that such an assumption is not uncommon in the OOD literature. From the AC's reading, I tend to agree that for the class of methods under consideration, it may be acceptable (even if, certainly, not fully desirable) to have a mild dependence on some additional OOD samples. The paper could benefit from a more upfront acknowledgement and discussion of this potential qualification for the applicability of the model.

Some other comments from the AC's reading:
- in the discussion of the sub-optimality of the MSP, it would be worth citing Bitterwolf et al., "Breaking Down Out-of-Distribution Detection", ICML 2022

- the papers refers to the complement set $\mathcal{Y}^{\rm c}$. What is the universe of all possible labels with respect to which the complement is taken?

- the paper refers to densities of the ID distribution. This requires some choice of base measure. This is typically Lebesgue; but does the precise choice play any role in the results?

- where does the density of the OOD distribution $D_{\rm Xo, Yo}$ feature in the analysis? Equation 2 appears to posit that regions of low ID density are OOD, as opposed to regions of low ID to OOD density *ratio*.

- a definition or reference for Legendre type functions would be appropriate.

**Justification For Why Not Higher Score:**

Technically solid work, but perhaps does not introduce a radically new idea for OOD detection.

**Justification For Why Not Lower Score:**

General agreement about the paper's novel theoretical contribution. Promising empirical results.

---

### Decision · Program_Chairs · 2024-01-16

Accept (poster)